# A PAC-Bayes Analysis of Adversarial Robustness

**Paul Viallard**[1*], **Guillaume Vidot**[23*], **Amaury Habrard**[1], **Emilie Morvant**[1]

[1] Univ Lyon, UJM-Saint-Etienne, CNRS, Institut d Optique Graduate School,
Laboratoire Hubert Curien UMR 5516, F-42023, SAINT-ETIENNE, France
[2] Airbus Opération S.A.S
[3] University of Toulouse, Institut de Recherche en Informatique de Toulouse, France

## Abstract

We propose the first general PAC-Bayesian generalization bounds for adversarial robustness, that estimate, at test time, how much a model will be invariant to imperceptible perturbations in the input. Instead of deriving a worst-case analysis of the risk of a hypothesis over all the possible perturbations, we leverage the PAC-Bayesian framework to bound the averaged risk on the perturbations for majority votes (over the whole class of hypotheses). Our theoretically founded analysis has the advantage to provide general bounds *(i)* that are valid for any kind of attacks (*i.e.*, the adversarial attacks), *(ii)* that are tight thanks to the PAC-Bayesian framework, *(iii)* that can be directly minimized during the learning phase to obtain a robust model on different attacks at test time.

## 1 Introduction

While machine learning algorithms are able to solve a huge variety of tasks, Szegedy et al. [2014] pointed out a crucial *weakness*: the possibility to generate samples similar to the originals (*i.e.*, with no or insignificant change recognizable by the human eyes) but with a different outcome from the algorithm. This phenomenon, known as "adversarial examples", contributes to the impossibility to ensure the safety of machine learning algorithms for safety-critical applications such as aeronautic functions (*e.g.*, vision-based navigation), autonomous driving, or medical diagnosis (see, *e.g.*, Huang et al. [2020]). Adversarial robustness is thus a critical issue in machine learning that studies the ability of a model to be robust or invariant to perturbations of its input. A perturbed input that fools the model is usually called an *adversarial example*. In other words, an adversarial example can be defined as an example that has been modified by an imperceptible noise (or that does not exceed a threshold) but which leads to a misclassification. One line of research is referred to as adversarial robustness verification [*e.g.*, Gehr et al., 2018, Huang et al., 2017, Singh et al., 2019, Tsuzuku et al., 2018], where the objective is to formally check whether the neighborhood of each sample does not contain any adversarial examples. This kind of method comes with some limitations such as scalability or overapproximation [Gehr et al., 2018, Katz et al., 2017, Singh et al., 2019]. In this paper we stand in another setting called adversarial attack/defense [*e.g.*, Papernot et al., 2016, Goodfellow et al., 2015, Madry et al., 2018, Carlini and Wagner, 2017, Zantedeschi et al., 2017, Kurakin et al., 2017]. An adversarial attack consists in finding perturbed examples that defeat machine learning algorithms while the adversarial defense techniques enhance their adversarial robustness to make the attacks useless. While a lot of methods exist, adversarial robustness suffers from a lack of general theoretical understandings (see Section 2.2).

To tackle this issue, we propose in this paper to formulate the adversarial robustness through the lens of a well-founded statistical machine learning theory called PAC-Bayes and introduced by Shawe-Taylor and Williamson [1997], McAllester [1998]. This theory has the advantage to provide tight

---

*Paul Viallard and Guillaume Vidot contributed equally to this work

35th Conference on Neural Information Processing Systems (NeurIPS 2021).

generalization bounds in average over the set of hypotheses considered (leading to bounds for a weighted majority vote over this set), in contrast to other theories such as VC-dimension or Rademacher-based approaches that give worst-case analysis, *i.e.*, for all the hypotheses. We start by defining our setting called *adversarially robust PAC-Bayes*. The idea consists in considering an *averaged adversarial robustness risk* which corresponds to the probability that the model misclassifies a perturbed example (this can be seen as an averaged risk over the perturbations). This measure can be too optimistic and not enough informative since for each example we sample only one perturbation. Thus we also define an *averaged-max adversarial risk* as the probability that there exists at least one perturbation (taken in a set of sampled perturbations) that leads to a misclassification. These definitions, based on averaged quantities, have the advantage *(i)* of still being suitable for the PAC-Bayesian framework and majority vote classifiers and *(ii)* to be related to the classical adversarial robustness risk. Then, for each of our adversarial risks, we derive a PAC-Bayesian generalization bound that can are valid to any kind of attack. From an algorithmic point of view, these bounds can be directly minimized in order to learn a majority vote robust in average to attacks. We empirically illustrate that our framework is able to provide generalization guarantees with non-vacuous bounds for the adversarial risk while ensuring efficient protection to adversarial attacks.

**Organization of the paper.** Section 2 recalls basics on usual adversarial robustness. We state our new adversarial robustness PAC-Bayesian setting along with our theoretical results in Section 3, and we empirically show its soundness in Section 4. All the proofs of the results are deferred in Appendix.

## 2    Basics on adversarial robustness

### 2.1    General setting

We tackle binary classification tasks with the input space $X=\mathbb{R}^d$ and the output/label space $Y=\{-1,+1\}$. We assume that $D$ is a fixed but unknown distribution on $X{\times}Y$. An example is denoted by $(x,y)\in X{\times}Y$. Let $S=\{(x_i,y_i)\}_{i=1}^m$ be the learning sample consisted of $m$ examples *i.i.d.* from $D$; We denote the distribution of such $m$-sample by $D^m$. Let $\mathcal{H}$ be a set of real-valued functions from $X$ to $[-1,+1]$ called voters or hypotheses. Usually, given a learning sample $S\sim D^m$, a learner aims at finding the best hypothesis $h$ from $\mathcal{H}$ that commits as few errors as possible on unseen data from $D$. One wants to find $h\in\mathcal{H}$ that minimizes the true risk $R_D(h)$ on $D$ defined as

$$R_D(h) = \mathop{\mathbb{E}}_{(x,y)\sim D} \ell\left(h,(x,y)\right), \tag{1}$$

where $\ell:\mathcal{H}{\times}X{\times}Y{\to}\mathbb{R}^+$ is the loss function. In practice since $D$ is unknown we cannot compute $R_D(h)$, we usually deal with the empirical risk $R_S(h)$ estimated on $S$ and defined as

$$R_S(h) = \frac{1}{m}\sum_{i=1}^m \ell(h,(x_i,y_i)).$$

From a classic ideal machine learning standpoint, we are able to learn a well-performing classifier with strong guarantees on unseen data, and even to measure how much the model will be able to generalize on $D$ (*e.g.*, with generalization bounds).

However, in real-life applications at classification time, an imperceptible perturbation of the input (*e.g.*, due to a malicious attack or a noise) can have a bad influence on the classification performance on unseen data [Szegedy et al., 2014]: the usual guarantees do not stand anymore. Such imperceptible perturbation can be modeled by a (relatively small) noise in the input. Let $b>0$ and $\|\cdot\|$ be an arbitrary norm (the most used norms are the $\ell_1$, $\ell_2$ and $\ell_\infty$-norms), the set of possible noises $B$ is defined by

$$B=\{\epsilon \in \mathbb{R}^d \mid \|\epsilon\| \leq b\}.$$

The learner aims to find an *adversarial robust* classifier that is robust in average to all noises in $B$ over $(x,y)\sim D$. More formally, one wants to minimize the adversarial robust true risk $R_D^{\text{ROB}}(h)$ defined as

$$R_D^{\text{ROB}}(h) = \mathop{\mathbb{E}}_{(x,y)\sim D} \max_{\epsilon\in B} \ell\left(h,(x{+}\epsilon,y)\right). \tag{2}$$

Similarly as in the classic setting, since $D$ is unknown, $R_D^{\text{ROB}}(h)$ cannot be directly computed, and then one usually deals with the empirical adversarial risk

$$R_S^{\text{ROB}}(h) = \frac{1}{m}\sum_{i=1}^m \max_{\epsilon\in B}\ell\left(h,(x_i{+}\epsilon,y_i)\right).$$

That being said, a learned classifier $h$ should be robust to *adversarial attacks* that aim at finding an *adversarial example* $x+\epsilon^*(x,y)$ to fool $h$ for given example $(x,y)$, where $\epsilon^*(x,y)$ is defined as

$$\epsilon^*(x,y) \in \operatorname{argmax}_{\epsilon \in B} \ \ell(h, (x+\epsilon, y)). \tag{3}$$

In consequence, *adversarial defense* mechanisms often rely on the adversarial attacks by replacing the original examples with the adversarial ones during the learning phase; This procedure is called adversarial training. Even if there are other defenses, adversarial training appears to be one of the most efficient defense mechanisms [Ren et al., 2020].

## 2.2 Related works

**Adversarial Attacks/Defenses.** Numerous methods[2] exist to solve–or approximate–the optimization of Equation (3). Among them, the Fast Gradient Sign Method (FGSM [Goodfellow et al. [2015]]) is an attack consisting in generating a noise $\epsilon$ in the direction of the gradient of the loss function with respect to the input $x$. Kurakin et al. [2017] introduced IFGSM, an iterative version of FGSM: at each iteration, one repeats FGSM and adds to $x$ a noise, that is the sign of the gradient of the loss with respect to $x$. Following the same principle as IFGSM, Madry et al. [2018] proposed a method based on Projected Gradient Descent (PGD) that includes a random initialization of $x$ before the optimization. Another technique known as the *Carlini and Wagner Attack* [Carlini and Wagner, 2017] aims at finding adversarial examples $x+\epsilon^*(x,y)$ that are as close as possible to the original $x$, *i.e.*, they want an attack being the most imperceptible as possible. However, producing such imperceptible perturbation leads to a high-running time in practice. Contrary to the most popular techniques that look for a model with a low adversarial robust risk (Equation (2)), our work stands in another line of research where the idea is to relax this worst-case risk measure by considering an *averaged* adversarial robust risk over the noises instead of a max-based formulation [see, *e.g.*, Zantedeschi et al., 2017, Hendrycks and Dietterich, 2019]. Our averaged formulation is introduced in the Section 3.

**Generalization Bounds.** Recently, few generalization bounds for adversarial robustness have been introduced [*e.g.* Khim and Loh, 2018, Yin et al., 2019, Montasser et al., 2019, 2020, Cohen et al., 2019, Salman et al., 2019]. Khim and Loh, and Yin et al.'s results are Rademacher complexity-based bounds. The former makes use of a surrogate of the adversarial risk; The latter provides bounds in the specific case of neural networks and linear classifiers, and involves an unavoidable polynomial dependence on the dimension of the input. Montasser et al. study robust PAC-learning for PAC-learnable classes with finite VC-dimension for unweighted majority votes that have been "robustified" with a boosting algorithm. However, their algorithm requires to consider all possible adversarial perturbations for each example which is intractable in practice, and their bound suffers also from a large constant as indicated at the end of the Montasser et al. [Theorem 3.1 2019]'s proof. Cohen et al. provide bounds that estimate what is the minimum noise to get an adversarial example (in the case of perturbations expressed as Gaussian noise) while our results give the probability to be fooled by an adversarial example. Salman et al. leverage Cohen et al.'s method and adversarial training in order to get tighter bounds. Moreover, Farnia et al. present margin-based bounds on the adversarial robust risk for specific neural networks and attacks (such as FGSM or PGD). While they made use of a classical PAC-Bayes bound, their result is not a PAC-Bayesian analysis and stands in the family of uniform-convergence bounds [see Nagarajan and Kolter, 2019, Ap. J for details]. In this paper, we provide PAC-Bayes bounds for general models expressed as majority votes, their bounds are thus not directly comparable to ours.

## 3 Adversarially robust PAC-Bayes

Although few theoretical results exist, the majority of works come either without theoretical guarantee or with very specific theoretical justifications. In the following, we aim at giving a different point of view on adversarial robustness based on the so-called PAC-Bayesian framework. By leveraging this framework, we derive a general generalization bound for adversarial robustness based on an averaged notion of risk that allows us to learn robust models at test time. We introduce below our new setting referred to as adversarially robust PAC-Bayes.

---

[2]The reader can refer to Ren et al. [2020] for a survey on adversarial attacks and defenses.

### 3.1 Adversarially robust majority vote

The PAC-Bayesian framework provides practical and theoretical tools to analyze majority vote classifiers. Assuming the voters set $\mathcal{H}$ and a learning sample $S$ as defined in Section 2, our goal is not anymore to learn one classifier from $\mathcal{H}$ but to learn a well-performing weighted combination of the voters involved in $\mathcal{H}$, the weights being modeled by a distribution $\mathcal{Q}$ on $\mathcal{H}$. This distribution is called the posterior distribution and is learned from $S$ given a prior distribution $\mathcal{P}$ on $\mathcal{H}$. The learned weighted combination is called a $\mathcal{Q}$-weighted majority vote and is defined by

$$\forall x \in X, \quad H_{\mathcal{Q}}(x) = \text{sign}\left[\mathbb{E}_{h \sim \mathcal{Q}} h(x)\right]. \tag{4}$$

In the rest of the paper, we consider the 0-1 loss function classically used for majority votes in PAC-Bayes and defined as $\ell(h,(x,y)) = \mathbf{I}(h(x) \neq y)$ with $\mathbf{I}(a)=1$ if $a$ is true, and 0 otherwise. In this context, the adversarial perturbation related to Equation (3) becomes

$$\epsilon^*(x,y) \in \text{argmax}_{\epsilon \in B} \mathbf{I}(H_{\mathcal{Q}}(x+\epsilon) \neq y). \tag{5}$$

Optimizing this problem is intractable due to the non-convexity of $H_{\mathcal{Q}}$ induced by the sign function. Note that the adversarial attacks of the literature (like PGD or IFGSM) aim at finding the optimal perturbation $\epsilon^*(x,y)$, but, in practice one considers an approximation of this perturbation.

Hence, instead of searching for the noise that maximizes the chance of fooling the algorithm, we propose to model the perturbation according to an example-dependent distribution. First let us define $\omega_{(x,y)}$ a distribution, on the set of possible noises $B$, that is dependent on an example $(x,y) \in X \times Y$. Then we denote as $\mathbf{D}$ the distribution on $(X \times Y) \times B$ defined as $\mathbf{D}((x,y),\epsilon) = D(x,y) \cdot \omega_{(x,y)}(\epsilon)$ which further permits to generate *perturbed examples*. To estimate our risks (defined below) for a given example $(x_i, y_i) \sim D$, we consider a set of $n$ perturbations sampled from $\omega_{(x_i,y_i)}$ denoted by $\mathcal{E}_i = \{\epsilon_j^i\}_{j=1}^n$. Then we consider as a learning set the $m \times n$-sample $\mathbf{S} = \{((x_i,y_i), \mathcal{E}_i)\}_{i=1}^m \in (X \times Y \times B^n)^m$. In other words, each $((x_i,y_i), \mathcal{E}_i) \in \mathbf{S}$ is sampled from a distribution that we denote by $\mathbf{D}^n$ such that

$$\mathbf{D}^n((x_i,y_i), \mathcal{E}_i) = D(x_i,y_i) \cdot \prod_{j=1}^n \omega_{(x_i,y_i)}(\epsilon_j^i).$$

Then, inspired by the works of Zantedeschi et al. [2017], Hendrycks and Dietterich [2019], we define our *robustness averaged adversarial risk* as follows.

**Definition 1** (Averaged Adversarial Risk). *For any distribution $\mathbf{D}$ on $(X \times Y) \times B$, for any distribution $\mathcal{Q}$ on $\mathcal{H}$, the averaged adversarial risk of $H_{\mathcal{Q}}$ is defined as*

$$R_{\mathbf{D}}(H_{\mathcal{Q}}) = \Pr_{((x,y),\epsilon) \sim \mathbf{D}} (H_{\mathcal{Q}}(x+\epsilon) \neq y)$$
$$= \mathbb{E}_{((x,y),\epsilon) \sim \mathbf{D}} \mathbf{I}(H_{\mathcal{Q}}(x+\epsilon) \neq y).$$

*The empirical averaged adversarial risk is computed on a $m \times n$-sample $\mathbf{S} = \{((x_i,y_i), \mathcal{E}_i)\}_{i=1}^m$ is*

$$R_{\mathbf{S}}(H_{\mathcal{Q}}) = \frac{1}{mn} \sum_{i=1}^m \sum_{j=1}^n \mathbf{I}(H_{\mathcal{Q}}(x_i+\epsilon_j^i) \neq y_i).$$

As we will show in Proposition 3, the risk $R_{\mathbf{D}}(H_{\mathcal{Q}})$ can considered optimistic regarding $\epsilon^*(x,y)$ of Equation (5). Indeed, instead of taking the $\epsilon$ maximizing the loss, a unique $\epsilon$ is drawn from a distribution. Hence, it can lead to a non-informative risk regarding the occurrence of adversarial examples. To overcome this, we propose an extension that we refer as *averaged-max adversarial risk*.

**Definition 2** (Averaged-Max Adversarial Risk). *For any distribution $\mathbf{D}$ on $(X \times Y) \times B$, for any distribution $\mathcal{Q}$ on $\mathcal{H}$, the averaged-max adversarial risk of $H_{\mathcal{Q}}$ is defined as*

$$A_{\mathbf{D}^n}(H_{\mathcal{Q}}) = \Pr_{((x,y),\mathcal{E}) \sim \mathbf{D}^n} (\exists \epsilon \in \mathcal{E}, H_{\mathcal{Q}}(x+\epsilon) \neq y).$$

*The empirical averaged-max adversarial risk computed on a $m \times n$-sample $\mathbf{S} = \{((x_i,y_i), \mathcal{E}_i)\}_{i=1}^m$ is*

$$A_{\mathbf{S}}(H_{\mathcal{Q}}) = \frac{1}{m} \sum_{i=1}^m \max_{\epsilon \in \mathcal{E}_i} \mathbf{I}(H_{\mathcal{Q}}(x_i + \epsilon) \neq y_i).$$

For an example $(x,y) \sim D$, instead of checking if one perturbed example $x+\epsilon$ is adversarial, we sample $n$ perturbed examples $x+\epsilon_1, \ldots, x+\epsilon_n$ and we check if at least one example is adversarial.

## 3.2 Relations between the adversarial risks

Proposition 3 below shows the intrinsic relationships between the classical adversarial risk $R_D^{\text{ROB}}(H_{\mathcal{Q}})$ and our two relaxations $R_{\mathbf{D}}(H_{\mathcal{Q}})$ and $A_{\mathbf{D}^n}(H_{\mathcal{Q}})$. In particular, Proposition 3 shows that the larger $n$, the number of perturbed examples, the higher is the chance to get an adversarial example and then to be close to the adversarial risk $R_D^{\text{ROB}}(H_{\mathcal{Q}})$.

**Proposition 3.** *For any distribution* $\mathbf{D}$ *on* $(X \times Y) \times B$, *for any distribution* $\mathcal{Q}$ *on* $\mathcal{H}$, *for any* $(n, n') \in \mathbb{N}^2$, *with* $n \geq n' \geq 1$, *we have*

$$R_{\mathbf{D}}(H_{\mathcal{Q}}) \ \leq \ A_{\mathbf{D}^{n'}}(H_{\mathcal{Q}}) \ \leq \ A_{\mathbf{D}^n}(H_{\mathcal{Q}}) \ \leq \ R_D^{\text{ROB}}(H_{\mathcal{Q}}). \tag{6}$$

The left-hand side of Equation (6) confirms that the averaged adversarial risk $R_{\mathbf{D}}(H_{\mathcal{Q}})$ is optimistic regarding the classical $R_D^{\text{ROB}}(H_{\mathcal{Q}})$. Proposition 4 estimates how close $R_{\mathbf{D}}(H_{\mathcal{Q}})$ can be to $R_D^{\text{ROB}}(H_{\mathcal{Q}})$.

**Proposition 4.** *For any distribution* $\mathbf{D}$ *on* $(X \times Y) \times B$, *for any distribution* $\mathcal{Q}$ *on* $\mathcal{H}$, *we have*

$$R_D^{\text{ROB}}(H_{\mathcal{Q}}) - \text{TV}(\Pi \| \Delta) \ \leq \ R_{\mathbf{D}}(H_{\mathcal{Q}}),$$

*where* $\Delta$ *and* $\Pi$ *are distributions on* $X \times Y$, *and* $\Delta(x', y')$, *respectively* $\Pi(x', y')$, *corresponds to the probability of drawing a perturbed example* $(x+\epsilon)$ *with* $((x, y), \epsilon) \sim \mathbf{D}$, *respectively an adversarial example* $(x+\epsilon^*(x, y), y)$ *with* $(x, y) \sim D$. *We have*

$$\Delta(x', y') = \Pr_{((x,y),\epsilon) \sim \mathbf{D}} [x+\epsilon=x', y=y'], \quad and \quad \Pi(x', y') = \Pr_{(x,y) \sim D} [x+\epsilon^*(x, y)=x', y=y'], \tag{7}$$

*and* $\text{TV}(\Pi \| \Delta) = \mathop{\mathbb{E}}_{(x',y') \sim \Delta} \dfrac{1}{2} \left| \dfrac{\Pi(x',y')}{\Delta(x',y')} - 1 \right|$, *is the Total Variation (TV) distance between* $\Pi$ *and* $\Delta$.

Note that $\epsilon^*(x, y)$ depends on $\mathcal{Q}$, and hence $\Pi$ depends on $\mathcal{Q}$. From Equation (7), $R_D^{\text{ROB}}(H_{\mathcal{Q}})$ and $R_{\mathbf{D}}(H_{\mathcal{Q}})$ can be rewritten (see Lemmas 8 and 9 in Appendix B) respectively with $\Delta$ and $\Pi$ as

$$R_{\mathbf{D}}(H_{\mathcal{Q}}) = \Pr_{(x',y') \sim \Delta} [H_{\mathcal{Q}}(x') \neq y'], \quad and \quad R_D^{\text{ROB}}(H_{\mathcal{Q}}) = \Pr_{(x',y') \sim \Pi} [H_{\mathcal{Q}}(x') \neq y'].$$

Finally, Propositions 3 and 4 relate the adversarial risk $R_{\mathbf{D}}(H_{\mathcal{Q}})$ to the "standard" adversarial risk $R_D^{\text{ROB}}(H_{\mathcal{Q}})$. Indeed, by merging the two propositions we obtain

$$R_D^{\text{ROB}}(H_{\mathcal{Q}}) - \text{TV}(\Pi \| \Delta) \ \leq \ R_{\mathbf{D}}(H_{\mathcal{Q}}) \ \leq \ A_{\mathbf{D}^n}(H_{\mathcal{Q}}) \ \leq \ R_D^{\text{ROB}}(H_{\mathcal{Q}}). \tag{8}$$

Hence, the smaller the TV distance $\text{TV}(\Pi \| \Delta)$, the closer the averaged adversarial risk $R_{\mathbf{D}}(H_{\mathcal{Q}})$ is from $R_D^{\text{ROB}}(H_{\mathcal{Q}})$ and the more probable an example $((x, y), \epsilon)$ sampled from $\mathbf{D}$ would be adversarial, *i.e.*, when our "averaged" adversarial example looks like a "specific" adversarial example. Moreover, Equation (8) justifies that the PAC-Bayesian point of view makes sense for adversarial learning with theoretical guarantees: the PAC-Bayesian guarantees we derive in the next section for our adversarial risks also give some guarantees on the "standard risk" $R_D^{\text{ROB}}(H_{\mathcal{Q}})$.

## 3.3 PAC-Bayesian bounds on the adversarially robust majority vote

First of all, since $R_{\mathbf{D}}(H_{\mathcal{Q}})$ and $A_{\mathbf{D}^n}(H_{\mathcal{Q}})$ risks are not differentiable due to the indicator function, we propose to use a common surrogate in PAC-Bayes (known as the Gibbs risk): instead of considering the risk of the $\mathcal{Q}$-weighted majority vote, we consider the expectation over $\mathcal{Q}$ of the individual risks of the voters involved in $\mathcal{H}$. In our case, we define the surrogates with the linear loss as

$$\overline{R_{\mathbf{D}}}(H_{\mathcal{Q}}) = \mathop{\mathbb{E}}_{((x,y),\epsilon) \sim \mathbf{D}} \frac{1}{2} \left[ 1 - y \mathop{\mathbb{E}}_{h \sim \mathcal{Q}} h(x+\epsilon) \right],$$

$$and \quad \overline{A_{\mathbf{D}^n}}(H_{\mathcal{Q}}) = \mathop{\mathbb{E}}_{((x,y),\boldsymbol{\mathcal{E}}) \sim \mathbf{D}^n} \frac{1}{2} \left[ 1 - \min_{\epsilon \in \boldsymbol{\mathcal{E}}} \left( y \mathop{\mathbb{E}}_{h \sim \mathcal{Q}} h(x+\epsilon) \right) \right].$$

The next theorem relates these surrogates to our risks, implying that a generalization bound for $\overline{R_{\mathbf{D}}}(H_{\mathcal{Q}})$, *resp.* for $\overline{A_{\mathbf{D}^n}}(H_{\mathcal{Q}})$, leads to a generalization bound for $R_{\mathbf{D}}(H_{\mathcal{Q}})$, *resp.* $A_{\mathbf{D}^n}(H_{\mathcal{Q}})$.

**Theorem 5.** *For any distributions* $\mathbf{D}$ *on* $(X \times Y) \times B$ *and* $\mathcal{Q}$ *on* $\mathcal{H}$, *for any* $n > 1$, *we have*

$$R_{\mathbf{D}}(H_{\mathcal{Q}}) \leq 2 \overline{R_{\mathbf{D}}}(H_{\mathcal{Q}}), \qquad and \qquad A_{\mathbf{D}^n}(H_{\mathcal{Q}}) \leq 2 \overline{A_{\mathbf{D}^n}}(H_{\mathcal{Q}}).$$

Theorem 6 below presents our PAC-Bayesian generalization bounds for $\overline{R_{\mathbf{D}}}(H_{\mathcal{Q}})$. Before that, it is important to mention that the empirical counterpart of $\overline{R_{\mathbf{D}}}(H_{\mathcal{Q}})$ is computed on $\mathbf{S}$ which is composed of non identically independently distributed samples, meaning that a "classical" proof technique is not applicable. The trick here is to make use of a result of Ralaivola et al. [2010] that provides a *chromatic PAC-Bayes bound*, *i.e.*, a bound which supports non-independent data.

**Theorem 6.** *For any distribution $\mathbf{D}$ on $(X \times Y) \times B$, for any set of voters $\mathcal{H}$, for any prior $\mathcal{P}$ on $\mathcal{H}$, for any $n$, with probability at least $1-\delta$ over $\mathbf{S}$, for all posteriors $\mathcal{Q}$ on $\mathcal{H}$, we have*

$$\mathrm{kl}(\overline{R_{\mathbf{S}}}(H_{\mathcal{Q}}) \| \overline{R_{\mathbf{D}}}(H_{\mathcal{Q}})) \leq \frac{1}{m}\left[\mathrm{KL}(\mathcal{Q}\|\mathcal{P}) + \ln\frac{m+1}{\delta}\right], \quad (9)$$

$$and \quad \overline{R_{\mathbf{D}}}(H_{\mathcal{Q}}) \leq \overline{R_{\mathbf{S}}}(H_{\mathcal{Q}}) + \sqrt{\frac{1}{2m}\left[\mathrm{KL}(\mathcal{Q}\|\mathcal{P}) + \ln\frac{m+1}{\delta}\right]}, \quad (10)$$

$$where \quad \overline{R_{\mathbf{S}}}(H_{\mathcal{Q}}) = \frac{1}{mn}\sum_{i=1}^{m}\sum_{j=1}^{n}\frac{1}{2}\left[1 - y_i \mathbb{E}_{h\sim\mathcal{Q}} h(x_i + \epsilon_j^i)\right],$$

$\mathrm{kl}(a\|b) = a\ln\frac{a}{b} + (1-a)\ln\frac{1-a}{1-b}$, and $\mathrm{KL}(\mathcal{Q}\|\mathcal{P}) = \mathbb{E}_{h\sim\mathcal{P}} \ln\frac{\mathcal{P}(h)}{\mathcal{Q}(h)}$ *the KL-divergence between $\mathcal{P}$ and $\mathcal{Q}$.*

Surprisingly, this theorem states bounds that do not depend on the number of perturbed examples $n$ but only on the number of original examples $m$. The reason is that the $n$ perturbed examples are inter-dependent (see the proof in Appendix). Note that Equation (9) is expressed as a Seeger [2002]'s bound and is tighter but less interpretable than Equation (10) expressed as a McAllester [1998]'s bound; These bounds involve the usual trade-off between the empirical risk $\overline{R_{\mathbf{S}}}(H_{\mathcal{Q}})$ and $\mathrm{KL}(\mathcal{Q}\|\mathcal{P})$.

We now state a generalization bound for $\overline{A_{\mathbf{D}^n}}(H_{\mathcal{Q}})$. Since this value involves a minimum term, we cannot use the same trick as for Theorem 6. To bypass this issue, we use the TV distance between two "artificial" distributions on $\mathcal{E}_i$. Given $((x_i, y_i), \mathcal{E}_i) \in \mathbf{S}$, let $\pi_i$ be an arbitrary distribution on $\mathcal{E}_i$, and given $h \in \mathcal{H}$, let $\rho_i^h$ be a Dirac distribution on $\mathcal{E}_i$ such that $\rho_i^h(\epsilon) = 1$ if $\epsilon = \mathrm{argmax}_{\epsilon\in\mathcal{E}_i} \frac{1}{2}\left[1 - y_i h(x_i + \epsilon)\right]$ (*i.e.*, if $\epsilon$ is maximizing the linear loss), and 0 otherwise.

**Theorem 7.** *For any distribution $\mathbf{D}$ on $(X \times Y) \times B$, for any set of voters $\mathcal{H}$, for any prior $\mathcal{P}$ on $\mathcal{H}$, for any $n$, with probability at least $1-\delta$ over $\mathbf{S}$, for all posteriors $\mathcal{Q}$ on $\mathcal{H}$, for all $i \in \{1, \ldots, m\}$, for all distributions $\pi_i$ on $\mathcal{E}_i$ independent from a voter $h \in \mathcal{H}$, we have*

$$\overline{A_{\mathbf{D}^n}}(H_{\mathcal{Q}}) \leq \frac{1}{m}\mathbb{E}_{h\sim\mathcal{Q}}\sum_{i=1}^{m}\max_{\epsilon\in\mathcal{E}_i}\frac{1}{2}(1 - y_i h(x_i+\epsilon)) + \sqrt{\frac{1}{2m}\left[\mathrm{KL}(\mathcal{Q}\|\mathcal{P}) + \ln\frac{2\sqrt{m}}{\delta}\right]} \quad (11)$$

$$\leq \overline{A_{\mathbf{S}}}(H_{\mathcal{Q}}) + \frac{1}{m}\sum_{i=1}^{m}\mathbb{E}_{h\sim\mathcal{Q}} \mathrm{TV}(\rho_i^h\|\pi_i) + \sqrt{\frac{1}{2m}\left[\mathrm{KL}(\mathcal{Q}\|\mathcal{P}) + \ln\frac{2\sqrt{m}}{\delta}\right]}, \quad (12)$$

*where* $\overline{A_{\mathbf{S}}}(H_{\mathcal{Q}}) = \frac{1}{m}\sum_{i=1}^{m}\frac{1}{2}\left[1 - \min_{\epsilon\in\mathcal{E}_i}\left(y_i \mathbb{E}_{h\sim\mathcal{Q}} h(x_i+\epsilon)\right)\right]$, *and* $\mathrm{TV}(\rho\|\pi) = \mathbb{E}_{\epsilon\sim\pi}\frac{1}{2}\left|\left[\frac{\rho(\epsilon)}{\pi(\epsilon)}\right] - 1\right|$.

To minimize the true average-max risk $\overline{A_{\mathbf{D}^n}}(H_{\mathcal{Q}})$ from Equation (11), we have to minimize a trade-off between $\mathrm{KL}(\mathcal{Q}\|\mathcal{P})$ (*i.e.*, how much the posterior weights are close to the prior ones) and the empirical risk $\frac{1}{m}\mathbb{E}_{h\sim\mathcal{Q}}\sum_{i=1}^{m}\max_{\epsilon\in\mathcal{E}_i}\frac{1}{2}(1 - y_i h(x_i+\epsilon))$. However, to compute the empirical risk, the loss for each voter and each perturbation has to be calculated and can be time-consuming. With Equation (12), we propose an alternative, which can be efficiently optimized using $\frac{1}{m}\sum_{i=1}^{m}\mathbb{E}_{h\sim\mathcal{Q}} \mathrm{TV}(\rho_i^h\|\pi_i)$ and the empirical average-max risk $\overline{A_{\mathbf{S}}}(H_{\mathcal{Q}})$. Intuitively, Equation (12) can be seen as a trade-off between the empirical risk, which reflects the robustness of the majority vote, and two penalization terms: the KL term and the TV term. The KL-divergence $\mathrm{KL}(\mathcal{Q}\|\mathcal{P})$ controls how much the posterior $\mathcal{Q}$ can differ from the prior ones $\mathcal{P}$. While the TV term $\mathbb{E}_h \mathrm{TV}(\rho_i^h\|\pi_i)$ controls the diversity of the voters, *i.e.*, the ability of the voters to be fooled on the same adversarial example. From an algorithmic view, an interesting behavior is that the bound of Equation (12) stands for all distributions $\pi_i$ on $\mathcal{E}_i$. This suggests that given $(x_i, y_i)$, we want to find $\pi_i$ minimizing $\mathbb{E}_{h\sim\mathcal{Q}} \mathrm{TV}(\rho_i^h\|\pi_i)$. Ideally,

this term tends to 0 when $\pi_i$ is close[3] to $\rho_i^h$ and all voters have their loss maximized by the same perturbation $\epsilon \in \boldsymbol{\mathcal{E}}_i$.

To learn a well-performing majority vote, one solution is to minimize the right-hand side of the bounds, meaning that we would like to find a good trade-off between a low empirical risk $\overline{R_{\mathbf{S}}}(H_{\mathcal{Q}})$ or $\overline{A_{\mathbf{S}}}(H_{\mathcal{Q}})$ and a low divergence between the prior weights and the learned posterior ones $\text{KL}(\mathcal{Q}\|\mathcal{P})$.

# 4 Experimental evaluation on differentiable decision trees

In this section, we illustrate the soundness of our framework in the context of differentiable decision trees learning. First of all, we describe our learning procedure designed from our theoretical results.

## 4.1 From the bounds to an algorithm

We consider a finite voters set $\mathcal{H}$ consisting of differentiable decision trees [Kontschieder et al., 2016] where each $h \in \mathcal{H}$ is parametrized by a weight vector $\boldsymbol{w}^h$. Inspired by Masegosa et al. [2020], we learn the decision trees of $\mathcal{H}$ and a data-dependent prior distribution $\mathcal{P}$ from a first learning set $\mathcal{S}'$ (independent from $\mathcal{S}$); This is a common approach in PAC-Bayes [Parrado-Hernández et al., 2012, Lever et al., 2013, Dziugaite and Roy, 2018, Dziugaite et al., 2021]. Then, the posterior distribution is learned from the second learning set $\mathcal{S}$ by minimizing the bounds. This means we need to minimize the risk and the KL-divergence term. Our two-step learning procedure is summarized in Algorithm 1.

**Step 1.** Starting from an initial prior $\mathcal{P}_0$ and an initial set of voters $\mathcal{H}_0$, where each voter $h$ is parametrized by a weight vector $\boldsymbol{w}_0^h$, the objective of this step is to construct the hypothesis set $\mathcal{H}$ and the prior distribution $\mathcal{P}$ to give as input to Step 2 for minimizing the bound. To do so, at each epoch $t$ of the Step 1, we learn from $\mathcal{S}'$ an "intermediate" prior $\mathcal{P}_t$ on an "intermediate" hypothesis set $\mathcal{H}_t$ consisting of voters $h$ parametrized by the weights $\boldsymbol{w}_t^h$; Note that the optimization in Line 9 is done with respect to $\boldsymbol{w}_t = \{\boldsymbol{w}_t^h\}_{h \in \mathcal{H}_t}$. At each iteration of the optimizer, from Lines 4 to 7, for each $(x, y)$ of the current batch $\mathbb{S}'$, we attack the majority vote $H_{\mathcal{P}_t}$ to obtain a perturbed example $x + \epsilon$. Then, in Lines 8 and 9, we perform a forward pass in the majority vote with the perturbed examples and update the weights $\boldsymbol{w}_t$ and the prior $\mathcal{P}_t$ according to the linear loss. To sum up, from Lines 11 to 20 at the end of Step 1, the prior $\mathcal{P}$ and the hypothesis set $\mathcal{H}$ constructed for Step 2 are the ones associated to the best epoch $t^* \in \{1, \ldots, T'\}$ that permits to minimize $\overline{R_{\mathcal{S}_t}}(H_{\mathcal{P}_t})$, where $\mathcal{S}_t = \{\texttt{attack}(x, y) \mid (x, y) \in \mathcal{S}\}$ is the perturbed set obtained by attacking the majority vote $H_{\mathcal{P}_t}$.

**Step 2.** Starting from the prior $\mathcal{P}$ on $\mathcal{H}$ and the learning set $\mathcal{S}$, we perform the same process as in Step 1 except that the considered objective function corresponds to the desired bound to optimize (Line 30, denoted $\texttt{B}(\cdot)$). For the sake of readability, we deferred in Appendix G the definition of $\texttt{B}(\cdot)$ for Equations (9) and (12). Note that the "intermediate" priors do not depend on $\mathcal{S}$, since they are learned from $\mathcal{S}'$: the bounds are then valid.

## 4.2 Experiments[4]

In this section, we empirically illustrate that our PAC-Bayesian framework for adversarial robustness is able to provide generalization guarantees with non-vacuous bounds for the adversarial risk.

**Setting.** We stand in a white-box setting meaning that the attacker knows the voters set $\mathcal{H}$, the prior distribution $\mathcal{P}$, and the posterior one $\mathcal{Q}$. We empirically study 2 attacks with the $\ell_2$-norm and $\ell_\infty$-norm: the Projected Gradient Descent (PGD, Madry et al. [2018]) and the iterative version of FGSM (IFGSM, Kurakin et al. [2017]). We fix the number of iterations at $k=20$ and the step size at $\frac{b}{k}$ for PGD and IFGSM (where $b=1$ for $\ell_2$-norm and $b=0.1$ for $\ell_\infty$-norm). One specificity of our setting is that we deal with the perturbation distribution $\omega_{(x,y)}$. We propose PGD$_U$ and IFGSM$_U$, two variants of PGD and IFGSM. To attack an example with PGD$_U$ or IFGSM$_U$ we proceed with the following steps: *(1)* We attack the prior majority vote $H_{\mathcal{P}}$ with the attack PGD or IFGSM: we will obtain a first perturbation $\epsilon'$ ; *(2)* We sample $n$ uniform noises $\eta_1, \ldots, \eta_n$ between $-10^{-2}$ and $+10^{-2}$ ; *(3)* We set

---

[3]Note that, since $\rho_i^h$ is a Dirac distribution, we have $\mathbb{E}_h \text{TV}(\rho_i^h\|\pi_i) = \frac{1}{2}\left[1 - \mathbb{E}_h \pi_i(\epsilon_h^*) + \mathbb{E}_h \sum_{\epsilon \neq \epsilon_h^*} \pi_i(\epsilon)\right]$, with $\epsilon_h^* = \text{argmax}_{\epsilon \in \boldsymbol{\mathcal{E}}_i} \frac{1}{2}\left[1 - y_i h(x_i + \epsilon)\right]$.

[4]The source code is available at https://github.com/paulviallard/NeurIPS21-PB-Robustness.

---
**Algorithm 1** Average Adversarial Training with Guarantee

---
**Require:** $\mathcal{S}, \mathcal{S}'$: disjoint learning sets – $T, T'$: number of epochs – $\mathcal{P}_0$: initial prior – $\mathcal{H}_0$ (with $\boldsymbol{w}_0$): initial hypothesis set – $\texttt{attack}(\cdot)$: the attack function – $\texttt{B}(\cdot)$: the objective function associated to a bound

**Step 1** – prior and voters' set construction | **Step 2** – bound minimization

1: **for** $t$ from 1 to $T'$ **do**
2:    $\mathcal{P}_t \leftarrow \mathcal{P}_{t-1}$ and $\mathcal{H}_t \leftarrow \mathcal{H}_{t-1}$ ($\boldsymbol{w}_t \leftarrow \boldsymbol{w}_{t-1}$)
3:    **for all** batches $\mathbb{S}'$ (from $\mathcal{S}'$) **do**
4:      **for all** $(x, y) \in \mathbb{S}'$ **do**
5:        $(x+\epsilon, y) \leftarrow \texttt{attack}(x, y)$
6:        $\mathbb{S}' \leftarrow (\mathbb{S}' \setminus \{(x, y)\}) \cup \{(x+\epsilon, y)\}$
7:      **end for**
8:      Update $\mathcal{P}_t$ with $\nabla_{\mathcal{P}_t} \overline{R_{\mathbb{S}'}}(H_{\mathcal{P}_t})$
9:      Update $\boldsymbol{w}_t$ with $\nabla_{\boldsymbol{w}_t} \overline{R_{\mathbb{S}'}}(H_{\mathcal{P}_t})$
10:    **end for**
11:    $\mathcal{S}_t \leftarrow \emptyset$
12:    **for all** $(x, y) \in \mathcal{S}$ **do**
13:      $(x+\epsilon, y) \leftarrow \texttt{attack}(x, y)$
14:      $\mathcal{S}_t \leftarrow \mathcal{S}_t \cup \{(x+\epsilon, y)\}$
15:    **end for**
16:    $t^* \leftarrow \operatorname{argmin}_{t' \in \{1, \ldots, t\}} \overline{R_{\mathcal{S}_{t'}}}(H_{\mathcal{P}_{t'}})$
17:    $\mathcal{P} \leftarrow \mathcal{P}_{t^*}$
18:    $\mathcal{H} \leftarrow \mathcal{H}_{t^*}$
19: **end for**
20: **return** $(\mathcal{P}, \mathcal{H})$

21: $(\mathcal{P}, \mathcal{H}) \leftarrow$ Output of **Step 1**
22: $\mathcal{Q}_0 \leftarrow \mathcal{P}$
23: **for** $t$ from 1 to $T$ **do**
24:    **for all** batches $\mathbb{S}$ (from $\mathcal{S}$) **do**
25:      $\mathcal{Q}_t \leftarrow \mathcal{Q}_{t-1}$
26:      **for all** $(x, y) \in \mathbb{S}$ **do**
27:        $(x+\epsilon, y) \leftarrow \texttt{attack}(x, y)$
28:        $\mathbb{S} \leftarrow (\mathbb{S} \setminus \{(x, y)\}) \cup \{(x+\epsilon, y)\}$
29:      **end for**
30:      Update $\mathcal{Q}_t$ with $\nabla_{\mathcal{Q}_t} \texttt{B}_{\mathbb{S}}(H_{\mathcal{Q}_t})$
31:    **end for**
32:    $\mathcal{S}_t \leftarrow \emptyset$
33:    **for all** $(x, y) \in \mathcal{S}$ **do**
34:      $(x+\epsilon, y) \leftarrow \texttt{attack}(x, y)$
35:      $\mathcal{S}_t \leftarrow \mathcal{S}_t \cup \{(x+\epsilon, y)\}$
36:    **end for**
37:    $t^* \leftarrow \operatorname{argmin}_{t' \in \{1, \ldots, t\}} \texttt{B}_{\mathcal{S}_{t'}}(H_{\mathcal{Q}_{t'}})$
38:    $\mathcal{Q} \leftarrow \mathcal{Q}_{t^*}$
39: **end for**
40: **return** $(\mathcal{Q}, \mathcal{H})$

---

the $i$-th perturbation as $\epsilon_i = \epsilon' + \eta_i$. Note that, for PGD$_\text{U}$ and IFGSM$_\text{U}$, after one attack we end up with $n{=}100$ perturbed examples. We set $n{=}1$ when these attacks are used as a defense mechanism in Algorithm 1. Indeed since the adversarial training is iterative, we do not need to sample numerous perturbations for each example: we sample a new perturbation each time the example is forwarded through the decision trees. We also consider a naive defense referred to as UNIF that only adds a noise uniformly such that the $\ell_p$-norm of the added noise is lower than $b$.

We study the following scenarios of defense/attack. These scenarios correspond to all the pairs (Defense, Attack) belonging to the set $\{$—, UNIF, PGD, IFGSM$\} \times \{$—, PGD, IFGSM$\}$ for the baseline, and $\{$—, UNIF, PGD$_\text{U}$, IFGSM$_\text{U}\} \times \{$—, PGD$_\text{U}$, IFGSM$_\text{U}\}$, where "—" means that we do not defend, *i.e.*, the attack returns the original example (note that PGD$_\text{U}$ and IFGSM$_\text{U}$ when "Attack without U" refers to PGD and IFGSM for computing the classical adversarial risk $R^{\text{ROB}}()$).

**Datasets and algorithm description.** We perform our experiment on six binary classification tasks from MNIST [LeCun et al., 1998] (1vs7, 4vs9, 5vs6) and Fashion MNIST [Xiao et al., 2017] (Coat vs Shirt, Sandal vs Ankle Boot, Top vs Pullover). We decompose the learning set into two disjoint subsets $\mathcal{S}'$ of around $7,000$ examples (to learn the prior and the voters) and $\mathcal{S}$ of exactly $5,000$ examples (to learn the posterior). We keep as test set $\mathcal{T}$ the original test set that contains around $2,000$ examples. Moreover, we need a perturbed test set, denoted by $\mathbf{T}$, to compute our averaged(-max) adversarial risks. Depending on the scenario, $\mathbf{T}$ is constructed from $\mathcal{T}$ by attacking the prior model $H_\mathcal{P}$ with PGD$_\text{U}$ or IFGSM$_\text{U}$ with $n{=}100$ (more details are given in Appendix). We run our Algorithm 1 for Equation (9) (Theorem 6), respectively Equation (12) (Theorem 7), and we compute our risk $R_\mathbf{T}(H_\mathcal{Q})$, respectively $A_\mathbf{T}(H_\mathcal{Q})$, the bound value and the usual adversarial risk associated to the model learned $R_\mathcal{T}^{\text{ROB}}(H_\mathcal{Q})$. Note that, during the evaluation of the bounds, we have to compute our relaxed adversarial risks $R_\mathbf{S}(H_\mathcal{Q})$ and $A_\mathbf{S}(H_\mathcal{Q})$ on $\mathcal{S}$. For Step 1, the initial prior $P_0$ is fixed to the uniform distribution, the initial set of voters $\mathcal{H}_0$ is constructed with weights initialized with Xavier Initializer [Glorot and Bengio, 2010] and bias initialized at 0 (more details are given in Appendix). During Step 2, to optimize the bound, we fix the confidence parameter $\delta{=}0.05$, and we consider as the set of voters $\mathcal{H}$ two settings: $\mathcal{H}$ as it is output by Step 1, and the set $\mathcal{H}^{\text{SIGN}} = \{h'(\cdot) = \text{sign}(h(\cdot)) \mid h \in \mathcal{H}\}$ for which the theoretical results are still valid (we will see that in this latter situation we are able to better minimize the TV term of Theorem 7). For the two steps, we use Adam optimizer [Kingma and Ba, 2015] for $T{=}T'{=}20$ epochs with a learning rate at $10^{-2}$ and a batch size at 64.

Table 1: Test risks and bounds for **MNIST 1vs7** with $n=100$ perturbations for all pairs (Defense,Attack) with the two voters' set $\mathcal{H}$ and $\mathcal{H}^{\text{SIGN}}$. The results in **bold** correspond to the best values between results for $\mathcal{H}$ and $\mathcal{H}^{\text{SIGN}}$. To quantify the gap between our risks and the classical definition we put in *italic* the risk of our models against the classical attacks: we replace $\text{PGD}_{\text{U}}$ and $\text{IFGSM}_{\text{U}}$ by PGD or IFGSM (*i.e.*, we did *not* sample from the uniform distribution). Since Eq. (12) upper-bounds Eq. (11) thanks to the TV term, we compute the two bound values of Theorem 7.

| $\ell_2$-norm $b=1$ | | Algo.1 with Eq. (9) | | | | | | Algo.1 with Eq. (12) | | | | | | | |
|---|---|---|---|---|---|---|---|---|---|---|---|---|---|---|---|
| | | Attack without U $R_{\mathcal{T}}^{\text{ROB}}(H_{\mathcal{Q}})$ | | $R_{\mathbf{T}}(H_{\mathcal{Q}})$ | | Th. 6 | | Attack without U $R_{\mathcal{T}}^{\text{ROB}}(H_{\mathcal{Q}})$ | | $A_{\mathbf{T}}(H_{\mathcal{Q}})$ | | Th. 7 - Eq. (12) | | Th. 7 - Eq. (11) | |
| Defense | Attack | $\mathcal{H}^{\text{SIGN}}$ | $\mathcal{H}$ | $\mathcal{H}^{\text{SIGN}}$ | $\mathcal{H}$ | $\mathcal{H}^{\text{SIGN}}$ | $\mathcal{H}$ | $\mathcal{H}^{\text{SIGN}}$ | $\mathcal{H}$ | $\mathcal{H}^{\text{SIGN}}$ | $\mathcal{H}$ | $\mathcal{H}^{\text{SIGN}}$ | $\mathcal{H}$ | $\mathcal{H}^{\text{SIGN}}$ | $\mathcal{H}$ |
| — | — | *.005* | *.005* | **.005** | .005 | **.017** | .019 | *.005* | *.005* | .005 | .005 | **.099** | 0.100 | **.099** | .100 |
| — | $\text{PGD}_{\text{U}}$ | **.245** | .255 | **.263** | .276 | .577 | **.448** | .315 | **.313** | **.325** | .326 | **.801** | 1.667 | .684 | **.515** |
| — | $\text{IFGSM}_{\text{U}}$ | **.084** | .086 | **.066** | .080 | **.170** | .185 | .117 | **.113** | **.106** | .110 | **.356** | 1.431 | .286 | **.251** |
| UNIF | — | *.005* | *.005* | .005 | .005 | **.018** | .019 | *.005* | *.005* | .005 | .005 | **.099** | 0.100 | **.099** | .100 |
| UNIF | $\text{PGD}_{\text{U}}$ | *.151* | **.146** | **.151** | .158 | .355 | **.292** | *.183* | **.178** | .190 | **.189** | **.531** | 1.620 | .454 | **.355** |
| UNIF | $\text{IFGSM}_{\text{U}}$ | *.063* | **.061** | **.031** | .035 | **.088** | .114 | *.071* | **.070** | .056 | **.054** | **.248** | 1.405 | .200 | **.186** |
| $\text{PGD}_{\text{U}}$ | — | **.006** | *.007* | **.006** | *.007* | **.023** | .024 | **.006** | *.007* | **.006** | .007 | **.102** | 0.103 | **.102** | .103 |
| $\text{PGD}_{\text{U}}$ | $\text{PGD}_{\text{U}}$ | **.028** | .030 | **.021** | .025 | .065 | **.064** | **.028** | .029 | **.025** | .028 | **.143** | 1.389 | .137 | **.136** |
| $\text{PGD}_{\text{U}}$ | $\text{IFGSM}_{\text{U}}$ | **.021** | .022 | **.013** | .016 | **.043** | .045 | *.022* | .022 | **.018** | .019 | **.125** | 1.362 | .121 | **.119** |
| $\text{IFGSM}_{\text{U}}$ | — | **.006** | *.007* | **.006** | *.007* | **.019** | .021 | **.006** | *.007* | **.006** | .007 | **.100** | 0.102 | **.100** | .102 |
| $\text{IFGSM}_{\text{U}}$ | $\text{PGD}_{\text{U}}$ | **.040** | .041 | **.033** | .035 | **.086** | .094 | *.040* | .039 | .040 | **.038** | **.184** | 1.368 | .166 | **.163** |
| $\text{IFGSM}_{\text{U}}$ | $\text{IFGSM}_{\text{U}}$ | **.021** | .022 | **.013** | .014 | **.039** | .049 | *.021* | .022 | **.018** | .021 | **.131** | 1.329 | **.122** | .123 |

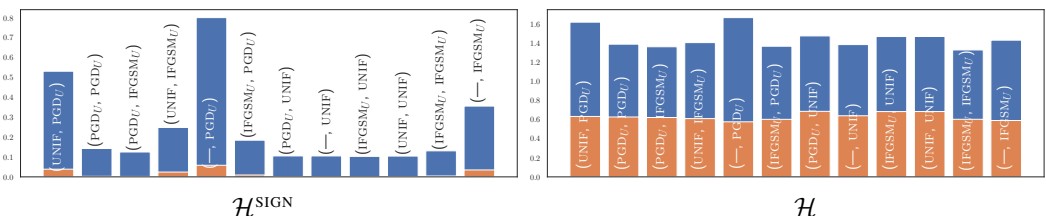

Figure 1: Visualization of the impact of the TV term in Equation (12). The left, respectively the right, bar plot show the bounds for the set of voters $\mathcal{H}^{\text{SIGN}}$, respectively $\mathcal{H}$. We plot the bounds for all the scenarios of Table 1 that use the TV distance, *i.e.*, all except the pairs $(\cdot, \text{—})$. In orange we represent the value of the TV term while in blue we represent all the remaining terms of the bound.

**Analysis of the results.** For the sake of readability, we exhibit the detailed results for one task (MNIST:1vs7) and all the pairs (Defense,Attack) with $\ell_2$-norm in Table 1, and we report in Figure 1 the influence of the TV term in the bound of Theorem 7 (Equation (12)). The detailed results on the other tasks are reported in Appendix; We provide in Figure 2 an overview of the results we obtained on all the tasks for the pairs (Defense,Attack) where "Defense=Attack" and with $\mathcal{H}^{\text{SIGN}}$.

First of all, from Table 1 the bounds of Theorem 6 are tighter than the ones of Theorem 7: this is an expected result since we showed that the averaged-max adversarial risk $A_{\mathbf{D}^n}(H_{\mathcal{Q}})$ is more pessimistic than its averaged counterpart $R_{\mathbf{D}}(H_{\mathcal{Q}})$. Note that the bound values of Equation (11) are tighter than the ones of Equation (12). This is expected since Equation (11) is a lower bound on Equation (12).

Second, the bounds with $\mathcal{H}^{\text{SIGN}}$ are all informative (lower than 1) and give insightful guarantees for our models. For Theorem 7 (Equation (12)) with $\mathcal{H}$, while the risks are comparable to the risks obtained with $\mathcal{H}^{\text{SIGN}}$, the bound values are greater than 1, meaning that we have no more guarantee on the model learned. As we can observe in Figure 1, this is due to the TV term involved in the bound. Considering $\mathcal{H}^{\text{SIGN}}$ when optimizing $A(\cdot)$ helps to control the TV term. Even if the bounds are non-vacuous for Theorem 6 with $\mathcal{H}$, the best models with the best guarantees are obtained with $\mathcal{H}^{\text{SIGN}}$. This is confirmed by the columns $R_{\mathcal{T}}^{\text{ROB}}(H_{\mathcal{Q}})$ that are always worse than $R_{\mathbf{T}}(H_{\mathcal{Q}})$ and mostly worse than $A_{\mathbf{T}}(H_{\mathcal{Q}})$ with $\mathcal{H}^{\text{SIGN}}$. The performance obtained with $\mathcal{H}^{\text{SIGN}}$ can be explained by the fact that the sign "saturates" the output of the voters which makes the majority vote more robust to noises. Thus, we focus the rest of the analysis on results obtained with $\mathcal{H}^{\text{SIGN}}$.

Third, we observe that the naive defense UNIF is able to improve the risks $R_{\mathbf{T}}(H_{\mathcal{Q}})$ and $A_{\mathbf{T}}(H_{\mathcal{Q}})$, but the improvement with the defenses based on $\text{PGD}_{\text{U}}$ and $\text{IFGSM}_{\text{U}}$ is much more significant specifically against a $\text{PGD}_{\text{U}}$ attack (up to 13 times better). We observe the same phenomenon for both bounds

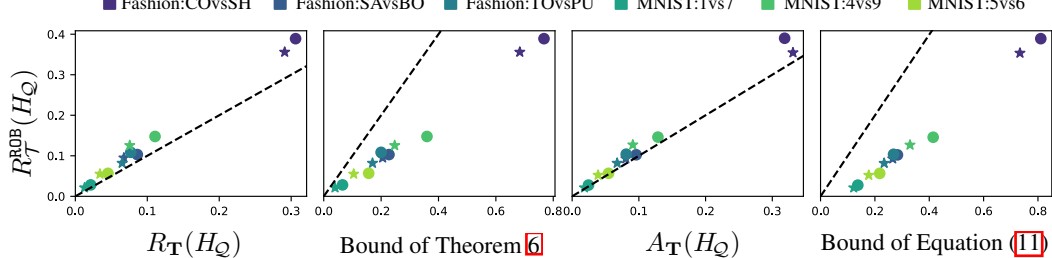

Figure 2: Visualization of the risk and bound values when "Defense=Attack" when the set of voters is $\mathcal{H}^{\mathrm{SIGN}}$. Results obtained with the $\mathrm{PGD_U}$, respectively $\mathrm{IFGSM_U}$, defense are represented by a star ★, respectively a circle ● *(reminder: $R_{\mathcal{T}}^{\mathrm{ROB}}(H_{\mathcal{Q}})$ is computed with a PGD, respectively IFGSM, attack).* The dashed line corresponds to bisecting line $y{=}x$. For $R_{\mathbf{T}}(H_{\mathcal{Q}})$ and $A_{\mathbf{T}}(H_{\mathcal{Q}})$, the closer the datasets are to the bisecting line, the more accurate our relaxed risk is compared to the classical adversarial risk $R_{\mathcal{T}}^{\mathrm{ROB}}(H_{\mathcal{Q}})$. For the bounds, the closer the datasets are to the bisecting line, the tighter the bound.

(Theorems 6 and 7). This is an interesting fact because this behavior confirms that we are able to learn models that are robust against the attacks tested with theoretical guarantees.

Lastly, from Figure 2 and Table 1, it is important to notice that the gap between the classical risk and our relaxed risks is small, meaning that our relaxation are not too optimistic. Despite the pessimism of the classical risk $R_{\mathcal{T}}^{\mathrm{ROB}}(H_{\mathcal{Q}})$, it remains consistent with our bounds, *i.e.*, it is lower than the bounds. In other words, in addition to giving upper bounds for our risks $R_{\mathbf{T}}(H_{\mathcal{Q}})$ and $A_{\mathbf{T}}(H_{\mathcal{Q}})$, our bounds give non-vacuous guarantees on the classical risks $R_{\mathcal{T}}^{\mathrm{ROB}}(H_{\mathcal{Q}})$.

## 5 Conclusion

To the best of our knowledge, our work is the first one that studies from a general standpoint adversarial robustness through the lens of the PAC-Bayesian framework. We have started by formalizing a new adversarial robustness setting (for binary classification) specialized for models that can be expressed as a weighted majority vote; we referred to this setting as Adversarially Robust PAC-Bayes. This formulation allowed us to derive PAC-Bayesian generalization bounds on the adversarial risk of general majority votes. We illustrated the usefulness of this setting on the training of (differentiable) decision trees. Our contribution is mainly theoretical and it does not appear to directly lead to potentially negative social impact.

This work gives rise to many interesting questions and lines of future research. Some perspectives will focus on extending our results to other classification settings such as multiclass or multilabel. Another line of research could focus on taking advantage of other tools of the PAC-Bayesian literature. Among them, we can make use of other bounds on the risk of the majority vote that take into consideration the diversity between the individual voters; For example, the C-bound [Lacasse et al., 2006], or more recently the tandem loss [Masegosa et al., 2020]. Another very recent PAC-Bayesian bound for majority votes that needs investigation in the case of adversarial robustness is the one proposed by Zantedeschi et al. [2021] that has the advantage to be directly optimizable with the 0-1 loss. Last but not least, in real-life applications, one often wants to combine different input sources (from different sensors, cameras, etc). Being able to combine these sources in an effective way is then a key issue. We believe that our new adversarial robustness setting can offer theoretical guarantees and well-founded algorithms when the model we learn is expressed as a majority vote, whether for ensemble methods with weak voters [*e.g.* Roy et al., 2011, Lorenzen et al., 2019], or for fusion of classifiers [*e.g.* Morvant et al., 2014], or for multimodal/multiview learning [*e.g.* Sun et al., 2017, Goyal et al., 2019].

## Acknowledgments and Disclosure of Funding

This work was partially funded supported by the French Project APRIORI ANR-18-CE23-0015. G. Vidot is supported by the ANRT with the convention "CIFRE" N°2019/0507. We also thank all anonymous reviewers for their constructive comments, and the time they took to review our work.

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
