| 17: $\quad \mathcal{P} \leftarrow \mathcal{P}_{t^*}$ | 37: $\quad t^* \leftarrow \operatorname{argmin}_{t' \in \{1,\dots,t\}} \texttt{B}_{\mathcal{S}_{t'}}(H_{\mathcal{Q}_{t'}})$ |
| 18: $\quad \mathcal{H} \leftarrow \mathcal{H}_{t^*}$ | 38: $\quad \mathcal{Q} \leftarrow \mathcal{Q}_{t^*}$ |
| 19: **end for** | 39: **end for** |
| 20: **return** $(\mathcal{P}, \mathcal{H})$ | 40: **return** $(\mathcal{Q}, \mathcal{H})$ |

---

the $i$-th perturbation as $\epsilon_i = \epsilon' + \eta_i$. Note that, for PGD$_\text{U}$ and IFGSM$_\text{U}$, after one attack we end up with $n{=}100$ perturbed examples. We set $n{=}1$ when these attacks are used as a defense mechanism in Algorithm 1. Indeed since the adversarial training is iterative, we do not need to sample numerous perturbations for each example: we sample a new perturbation each time the example is forwarded through the decision trees. We also consider a naive defense referred to as UNIF that only adds a noise uniformly such that the $\ell_p$-norm of the added noise is lower than $b$.

We study the following scenarios of defense/attack. These scenarios correspond to all the pairs (Defense, Attack) belonging to the set $\{\text{—}, \text{UNIF}, \text{PGD}, \text{IFGSM}\} \times \{\text{—}, \text{PGD}, \text{IFGSM}\}$ for the baseline, and $\{\text{—}, \text{UNIF}, \text{PGD}_\text{U}, \text{IFGSM}_\text{U}\} \times \{\text{—}, \text{PGD}_\text{U}, \text{IFGSM}_\text{U}\}$, where "—" means that we do not defend, *i.e.*, the attack returns the original example (note that PGD$_\text{U}$ and IFGSM$_\text{U}$ when "Attack without U" refers to PGD and IFGSM for computing the classical adversarial risk $R^{\text{ROB}}()$).

**Datasets and algorithm description.** We perform our experiment on six binary classification tasks from MNIST [LeCun et al., 1998] (1vs7, 4vs9, 5vs6) and Fashion MNIST [Xiao et al., 2017] (Coat vs Shirt, Sandal vs Ankle Boot, Top vs Pullover). We decompose the learning set into two disjoint subsets $\mathcal{S}'$ of around $7,000$ examples (to learn the prior and the voters) and $\mathcal{S}$ of exactly $5,000$ examples (to learn the posterior). We keep as test set $\mathcal{T}$ the original test set that contains around $2,000$ examples. Moreover, we need a perturbed test set, denoted by $\mathbf{T}$, to compute our averaged(-max) adversarial risks. Depending on the scenario, $\mathbf{T}$ is constructed from $\mathcal{T}$ by attacking the prior model $H_\mathcal{P}$ with PGD$_\text{U}$ or IFGSM$_\text{U}$ with $n{=}100$ (more details are given in Appendix). We run our Algorithm 1 for Equation (9) (Theorem 6), respectively Equation (12) (Theorem 7), and we compute our risk $R_\mathbf{T}(H_\mathcal{Q})$, respectively $A_\mathbf{T}(H_\mathcal{Q})$, the bound value and the usual adversarial risk associated to the model learned $R_\mathcal{T}^{\text{ROB}}(H_\mathcal{Q})$. Note that, during the evaluation of the bounds, we have to compute our relaxed adversarial risks $R_\mathbf{S}(H_\mathcal{Q})$ and $A_\mathbf{S}(H_\mathcal{Q})$ on $\mathcal{S}$. For Step 1, the initial prior $P_0$ is fixed to the uniform distribution, the initial set of voters $\mathcal{H}_0$ is constructed with weights initialized with Xavier Initializer [Glorot and Bengio, 2010] and bias initialized at $0$ (more details are given in Appendix). During Step 2, to optimize the bound, we fix the confidence parameter $\delta{=}0.05$, and we consider as the set of voters $\mathcal{H}$ two settings: $\mathcal{H}$ as it is output by Step 1, and the set $\mathcal{H}^{\text{SIGN}} = \{h'(\cdot) = \text{sign}(h(\cdot)) \mid h \in \mathcal{H}\}$ for which the theoretical results are still valid (we will see that in this latter situation we are able to better minimize the TV term of Theorem 7). For the two steps, we use Adam optimizer [Kingma and Ba, 2015] for $T{=}T'{=}20$ epochs with a learning rate at $10^{-2}$ and a batch size at $64$.

Table 1: Test risks and bounds for **MNIST 1vs7** with $n=100$ perturbations for all pairs (Defense,Attack) with the two voters' set $\mathcal{H}$ and $\mathcal{H}^{\text{SIGN}}$. The results in **bold** correspond to the best values between results for $\mathcal{H}$ and $\mathcal{H}^{\text{SIGN}}$. To quantify the gap between our risks and the classical definition we put in *italic* the risk of our models against the classical attacks: we replace PGD$_\text{U}$ and IFGSM$_\text{U}$ by PGD or IFGSM (*i.e.*, we did *not* sample from the uniform distribution). Since Eq. (12) upper-bounds Eq. (11) thanks to the TV term, we compute the two bound values of Theorem 7.

| $\ell_2$-norm $b=1$ | | Algo.1 with Eq. (9) | | | | | | Algo.1 with Eq. (12) | | | | | | | |
|---|---|---|---|---|---|---|---|---|---|---|---|---|---|---|---|
| | | Attack without U $R_{\mathcal{T}}^{\text{ROB}}(H_Q)$ | | $R_{\mathbf{T}}(H_Q)$ | | Th. 6 | | Attack without U $R_{\mathcal{T}}^{\text{ROB}}(H_Q)$ | | $A_{\mathbf{T}}(H_Q)$ | | Th. 7 - Eq. (12) | | Th. 7 - Eq. (11) | |
| Defense | Attack | $\mathcal{H}^{\text{SIGN}}$ | $\mathcal{H}$ | $\mathcal{H}^{\text{SIGN}}$ | $\mathcal{H}$ | $\mathcal{H}^{\text{SIGN}}$ | $\mathcal{H}$ | $\mathcal{H}^{\text{SIGN}}$ | $\mathcal{H}$ | $\mathcal{H}^{\text{SIGN}}$ | $\mathcal{H}$ | $\mathcal{H}^{\text{SIGN}}$ | $\mathcal{H}$ | $\mathcal{H}^{\text{SIGN}}$ | $\mathcal{H}$ |
| — | — | *.005* | *.005* | .005 | .005 | **.017** | .019 | *.005* | *.005* | .005 | .005 | **.099** | 0.100 | **.099** | .100 |
| — | PGD$_\text{U}$ | **.245** | .255 | **.263** | .276 | .577 | **.448** | .315 | **.313** | **.325** | .326 | **.801** | 1.667 | .684 | **.515** |
| — | IFGSM$_\text{U}$ | **.084** | .086 | **.066** | .080 | **.170** | .185 | .117 | **.113** | **.106** | .110 | **.356** | 1.431 | .286 | **.251** |
| UNIF | — | *.005* | *.005* | .005 | .005 | **.018** | .019 | *.005* | *.005* | .005 | .005 | **.099** | 0.100 | **.099** | .100 |
| UNIF | PGD$_\text{U}$ | *.151* | ***.146*** | **.151** | .158 | .355 | **.292** | .183 | **.178** | .190 | **.189** | **.531** | 1.620 | .454 | **.355** |
| UNIF | IFGSM$_\text{U}$ | *.063* | ***.061*** | **.031** | .035 | **.088** | .114 | .071 | **.070** | .056 | **.054** | **.248** | 1.405 | .200 | **.186** |
| PGD$_\text{U}$ | — | ***.006*** | *.007* | **.006** | .007 | **.023** | .024 | ***.006*** | *.007* | **.006** | .007 | **.102** | 0.103 | **.102** | .103 |
| PGD$_\text{U}$ | PGD$_\text{U}$ | ***.028*** | *.030* | **.021** | .025 | .065 | **.064** | ***.028*** | *.029* | **.025** | .028 | **.143** | 1.389 | .137 | **.136** |
| PGD$_\text{U}$ | IFGSM$_\text{U}$ | ***.021*** | *.022* | **.013** | .016 | **.043** | .045 | *.022* | *.022* | **.018** | .019 | **.125** | 1.362 | .121 | **.119** |
| IFGSM$_\text{U}$ | — | ***.006*** | *.007* | **.006** | .007 | **.019** | .021 | ***.006*** | *.007* | **.006** | .007 | **.100** | 0.102 | **.100** | .102 |
| IFGSM$_\text{U}$ | PGD$_\text{U}$ | ***.040*** | *.041* | **.033** | .035 | .086 | **.094** | *.040* | *.039* | .040 | .038 | **.184** | 1.368 | .166 | **.163** |

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

# A PAC-Bayes Analysis of Adversarial Robustness
## Supplementary Material

The supplementary material is structured as follows. The sections from A to E are devoted to our proofs. We give details on Algorithm 1 and on the computation of the bounds in Section G. We discuss, in Section F, the validity of the bound when we select a prior with $\mathcal{S}$ and have a distribution on perturbations depending on this selected prior. We introduce, in Section H, the voters that we use in our majority vote. Finally, we present additional experiments in Section I.

## A    Proof of Proposition 3

**Proposition 3.** *For any distribution* $\mathbf{D}$ *on* $(X{\times}Y){\times}B$, *for any distribution* $\mathcal{Q}$ *on* $\mathcal{H}$, *for any* $(n, n') \in \mathbb{N}^2$, *with* $n \geq n' \geq 1$, *we have*

$$R_{\mathbf{D}}(H_{\mathcal{Q}}) \ \leq \ A_{\mathbf{D}^{n'}}(H_{\mathcal{Q}}) \ \leq \ A_{\mathbf{D}^{n}}(H_{\mathcal{Q}}) \ \leq \ R_D^{\text{ROB}}(H_{\mathcal{Q}}). \tag{13}$$

*For any distribution* $\mathbf{D}$ *on* $(X{\times}Y){\times}B$, *for any distribution* $\mathcal{Q}$ *on* $\mathcal{H}$, *for any* $(n, n') \in \mathbb{N}^2$, *with* $n \geq n' \geq 1$, *we have*

$$R_{\mathbf{D}}(H_{\mathcal{Q}}) \leq A_{\mathbf{D}^{n'}}(H_{\mathcal{Q}}) \leq A_{\mathbf{D}^{n}}(H_{\mathcal{Q}}) \leq R_D^{\text{ROB}}(H_{\mathcal{Q}}).$$

*Proof.* First, we prove $A_{\mathbf{D}^1}(H_{\mathcal{Q}}){=}R_{\mathbf{D}}(H_{\mathcal{Q}})$. We have

$$\begin{aligned}
A_{\mathbf{D}^1}(H_{\mathcal{Q}}) &= 1 - \Pr_{((x,y),\boldsymbol{\mathcal{E}})\sim\mathbf{D}^1} \left(\forall\epsilon \in \boldsymbol{\mathcal{E}}, H_{\mathcal{Q}}(x+\epsilon) = y\right) \\
&= 1 - \Pr_{((x,y),\boldsymbol{\mathcal{E}})\sim\mathbf{D}^1} \left(\forall\epsilon \in \{\epsilon_1\}, H_{\mathcal{Q}}(x+\epsilon) = y\right) \\
&= 1 - \Pr_{((x,y),\boldsymbol{\mathcal{E}})\sim\mathbf{D}^1} \left(H_{\mathcal{Q}}(x+\epsilon_1) = y\right) = R_{\mathbf{D}}(H_{\mathcal{Q}}).
\end{aligned}$$

Then, we prove the inequality $A_{\mathbf{D}^{n'}}(H_{\mathcal{Q}}) \leq A_{\mathbf{D}^{n}}(H_{\mathcal{Q}})$ from the fact that the indicator function $\mathbf{I}(\cdot)$ is upper-bounded by 1. Indeed, from Definition 2 we have

$$\begin{aligned}
1 - A_{\mathbf{D}^{n}}(H_{\mathcal{Q}}) &= \mathbb{E}_{(x,y)\sim D} \ \mathbb{E}_{\boldsymbol{\mathcal{E}}\sim\omega_{(x,y)}^n} \ \mathbf{I}\left(\forall\epsilon \in \boldsymbol{\mathcal{E}}, H_{\mathcal{Q}}(x+\epsilon) = y\right) \\
&= \mathbb{E}_{(x,y)\sim D} \left[\prod_{i=1}^{n} \ \mathbb{E}_{\epsilon_i\sim\omega_{(x,y)}} \mathbf{I}\left(H_{\mathcal{Q}}(x+\epsilon_i) = y\right)\right] \\
&\leq \mathbb{E}_{(x,y)\sim D} \left[\prod_{i=1}^{n'} \ \mathbb{E}_{\epsilon_i\sim\omega_{(x,y)}} \mathbf{I}\left(H_{\mathcal{Q}}(x+\epsilon_i) = y\right)\right] \\
&= \mathbb{E}_{(x,y)\sim D} \ \mathbb{E}_{\boldsymbol{\mathcal{E}}'\sim\omega_{(x,y)}^{n'}} \ \mathbf{I}\left(\forall\epsilon \in \boldsymbol{\mathcal{E}}', H_{\mathcal{Q}}(x+\epsilon) = y\right) \\
&= 1 - A_{\mathbf{D}^{n'}}(H_{\mathcal{Q}}).
\end{aligned}$$

Lastly, to prove the rightmost inequality, we have to use the fact that the expectation over the set $B$ is bounded by the maximum over the set $B$. We have

$$\begin{aligned}
A_{\mathbf{D}^{n}}(H_{\mathcal{Q}}) &= \mathbb{E}_{(x,y)\sim D} \ \mathbb{E}_{\epsilon_1\sim\omega_{(x,y)}} \cdots \mathbb{E}_{\epsilon_n\sim\omega_{(x,y)}} \mathbf{I}\left(\exists\epsilon{\in}\{\epsilon_1,\dots,\epsilon_n\}, H_{\mathcal{Q}}(x+\epsilon) \neq y\right) \\
&\leq \mathbb{E}_{(x,y)\sim D} \ \max_{\epsilon_1\in B} \cdots \max_{\epsilon_n\in B} \mathbf{I}\left(\exists\epsilon \in \{\epsilon_1,\dots,\epsilon_n\}, H_{\mathcal{Q}}(x+\epsilon) \neq y\right) \\
&= \mathbb{E}_{(x,y)\sim D} \ \max_{\epsilon_1\in B} \cdots \max_{\epsilon_{n-1}\in B} \mathbf{I}\left(\exists\epsilon \in \{\epsilon_1,\dots,\epsilon^*\}, H_{\mathcal{Q}}(x+\epsilon) \neq y\right) \\
&= \mathbb{E}_{(x,y)\sim D} \ \mathbf{I}\left(H_{\mathcal{Q}}(x+\epsilon^*) \neq y\right) \\
&= \mathbb{E}_{(x,y)\sim D} \ \max_{\epsilon\in B} \mathbf{I}\left(H_{\mathcal{Q}}(x+\epsilon) \neq y\right) \ = \ R_D^{\text{ROB}}(H_{\mathcal{Q}}).
\end{aligned}$$

Merging the three equations proves the claim. $\qquad\square$

## B  Proof of Proposition 4

In this section, we provide the proof of Proposition 4 that relies on Lemmas 8 and 9 which are also described and proved. Lemma 8 shows that $R_{\mathbf{D}}(H_{\mathcal{Q}})$ is equivalent to $R_{\Delta}(H_{\mathcal{Q}})$.

**Lemma 8.** *For any distribution $\mathbf{D}$ on $(X \times Y) \times B$ and its associated distribution $\Delta$, for any posterior $\mathcal{Q}$ on $\mathcal{H}$, we have*

$$R_{\mathbf{D}}(H_{\mathcal{Q}}) = \Pr_{(x+\epsilon,y)\sim\Delta}[H_{\mathcal{Q}}(x+\epsilon)\neq y] = R_{\Delta}(H_{\mathcal{Q}}).$$

*Proof.* Starting from the averaged adversarial risk $R_{\mathbf{D}}(H_{\mathcal{Q}}) = \mathbb{E}_{((x,y),\epsilon)\sim\mathbf{D}}\,\mathbf{I}[H_{\mathcal{Q}}(x+\epsilon)\neq y]$, we have

$$R_{\mathbf{D}}(H_{\mathcal{Q}}) = \mathbb{E}_{(x'+\epsilon',y')\sim\Delta}\frac{1}{\Delta(x'+\epsilon',y')}\left[\Pr_{((x,y),\epsilon)\sim\mathbf{D}}[H_{\mathcal{Q}}(x+\epsilon)\neq y,\,x'+\epsilon'=x+\epsilon,\,y'=y]\right]$$

$$= \mathbb{E}_{(x'+\epsilon',y')\sim\Delta}\frac{1}{\Delta(x'+\epsilon',y')}\left[\mathbb{E}_{((x,y),\epsilon)\sim\mathbf{D}}\mathbf{I}[H_{\mathcal{Q}}(x+\epsilon)\neq y]\,\mathbf{I}[x'+\epsilon'=x+\epsilon,\,y'=y]\right].$$

In other words, the double expectation only rearranges the terms of the original expectation: given an example $(x'+\epsilon',y')$, we gather probabilities such that $H_{\mathcal{Q}}(x+\epsilon)\neq y$ with $(x+\epsilon,y)=(x'+\epsilon',y')$ in the inner expectation, while integrating over all couple $(x'+\epsilon',y') \in X \times Y$ in the outer expectation. Then, from the fact that when $x'+\epsilon'=x+\epsilon$ and $y'=y$, $\mathbf{I}[H_{\mathcal{Q}}(x+\epsilon)\neq y] = \mathbf{I}[H_{\mathcal{Q}}(x'+\epsilon')\neq y']$, we have

$$R_{\mathbf{D}}(H_{\mathcal{Q}}) = \mathbb{E}_{(x'+\epsilon',y')\sim\Delta}\frac{1}{\Delta(x'+\epsilon',y')}\left[\mathbb{E}_{((x,y),\epsilon)\sim\mathbf{D}}\mathbf{I}[H_{\mathcal{Q}}(x'+\epsilon')\neq y']\mathbf{I}[x'+\epsilon'=x+\epsilon,\,y'=y]\right]$$

$$= \mathbb{E}_{(x'+\epsilon',y')\sim\Delta}\frac{1}{\Delta(x'+\epsilon',y')}\left[\mathbf{I}[H_{\mathcal{Q}}(x'+\epsilon')\neq y']\mathbb{E}_{((x,y),\epsilon)\sim\mathbf{D}}\mathbf{I}[x'+\epsilon'=x+\epsilon,\,y'=y]\right].$$

Finally, by definition of $\Delta(x'+\epsilon',y')$, we can deduce that

$$R_{\mathbf{D}}(H_{\mathcal{Q}}) = \mathbb{E}_{(x'+\epsilon',y')\sim\Delta}\frac{1}{\Delta(x'+\epsilon',y')}[\mathbf{I}[H_{\mathcal{Q}}(x'+\epsilon')\neq y']\,\Delta(x'+\epsilon',y')]$$

$$= \mathbb{E}_{(x'+\epsilon',y')\sim\Delta}\mathbf{I}[H_{\mathcal{Q}}(x'+\epsilon')\neq y'] = R_{\Delta}(H_{\mathcal{Q}}).$$

$\square$

Similarly, Lemma 9 shows that $R_D^{\text{ROB}}(H_{\mathcal{Q}})$ is equivalent to $R_{\Pi}(H_{\mathcal{Q}})$.

**Lemma 9.** *For any distribution $D$ on $X \times Y$ and its associated distribution $\Pi$, for any posterior $\mathcal{Q}$ on $\mathcal{H}$, we have*

$$R_D^{\text{ROB}}(H_{\mathcal{Q}}) = \Pr_{(x+\epsilon,y)\sim\Pi}[H_{\mathcal{Q}}(x+\epsilon)\neq y] = R_{\Pi}(H_{\mathcal{Q}}).$$

*Proof.* The proof is similar to the one of Lemma 8. Indeed, starting from the definition of $R_D^{\text{ROB}}(H_{\mathcal{Q}}) = \mathbb{E}_{(x,y)\sim D}\,\mathbf{I}[H_{\mathcal{Q}}(x+\epsilon^*(x,y)) \neq y]$, we have

$$R_D^{\text{ROB}}(H_{\mathcal{Q}}) = \mathbb{E}_{(x'+\epsilon',y')\sim\Pi}\frac{1}{\Pi(x'+\epsilon',y')}\left[\mathbb{E}_{(x,y)\sim D}\mathbf{I}\left[H_{\mathcal{Q}}(x+\epsilon^*(x,y)) \neq y\right]\mathbf{I}[x'+\epsilon'=x+\epsilon^*(x,y),\,y'=y]\right]$$

$$= \mathbb{E}_{(x'+\epsilon',y')\sim\Pi}\frac{1}{\Pi(x'+\epsilon',y')}\left[\mathbb{E}_{(x,y)\sim D}\mathbf{I}\left[H_{\mathcal{Q}}(x'+\epsilon') \neq y'\right]\mathbf{I}[x'+\epsilon'=x+\epsilon^*(x,y),\,y'=y]\right].$$

Finally, by definition of $\Pi(x'+\epsilon',y')$, we can deduce that

$$R_D^{\text{ROB}}(H_{\mathcal{Q}}) = \mathbb{E}_{(x'+\epsilon',y')\sim\Pi}\frac{1}{\Pi(x'+\epsilon',y')}\left[\mathbf{I}\left[H_{\mathcal{Q}}(x'+\epsilon') \neq y'\right]\Pi(x'+\epsilon',y')\right]$$

$$= \mathbb{E}_{(x'+\epsilon',y')\sim\Pi}\mathbf{I}\left[H_{\mathcal{Q}}(x'+\epsilon')\neq y'\right] = R_{\Pi}(H_{\mathcal{Q}}).$$

$\square$

We can now prove Proposition 4.

**Proposition 4.** *For any distribution* $\mathbf{D}$ *on* $(X \times Y) \times B$, *for any distribution* $\mathcal{Q}$ *on* $\mathcal{H}$, *we have*

$$R_D^{\text{ROB}}(H_{\mathcal{Q}}) - \text{TV}(\Pi \| \Delta) \leq R_{\mathbf{D}}(H_{\mathcal{Q}}).$$

*Proof.* From Lemmas 8 and 9, we have

$$R_{\mathbf{D}}(H_{\mathcal{Q}}) = R_{\Delta}(H_{\mathcal{Q}}), \quad \text{and} \quad R_D^{\text{ROB}}(H_{\mathcal{Q}}) = R_{\Pi}(H_{\mathcal{Q}}).$$

Then, we apply Lemma 4 of Ohnishi and Honorio [2021], we have

$$R_{\Pi}(H_{\mathcal{Q}}) \leq \text{TV}(\Pi \| \Delta) + R_{\Delta}(H_{\mathcal{Q}}) \quad \Longleftrightarrow \quad R_D^{\text{ROB}}(H_{\mathcal{Q}}) \leq \text{TV}(\Pi \| \Delta) + R_{\mathbf{D}}(H_{\mathcal{Q}}).$$

$\square$

# C Proof of Theorem 5

**Theorem 5.** *For any distributions* $\mathbf{D}$ *on* $(X \times Y) \times B$ *and* $\mathcal{Q}$ *on* $\mathcal{H}$, *for any* $n > 1$, *we have*

$$R_{\mathbf{D}}(H_{\mathcal{Q}}) \leq 2\overline{R_{\mathbf{D}}}(H_{\mathcal{Q}}), \quad \text{and} \quad A_{\mathbf{D}^n}(H_{\mathcal{Q}}) \leq 2\overline{A_{\mathbf{D}^n}}(H_{\mathcal{Q}}).$$

*Proof.* By the definition of the majority vote, we have

$$\frac{1}{2}R_{\mathbf{D}}(H_{\mathcal{Q}}) = \frac{1}{2} \Pr_{((x,y),\epsilon) \sim \mathbf{D}} \left( y \mathop{\mathbb{E}}_{h \sim \mathcal{Q}} h(x+\epsilon) \leq 0 \right)$$

$$= \frac{1}{2} \Pr_{((x,y),\epsilon) \sim \mathbf{D}} \left( 1 - y \mathop{\mathbb{E}}_{h \sim \mathcal{Q}} h(x+\epsilon) \geq 1 \right)$$

$$\leq \mathop{\mathbb{E}}_{((x,y),\epsilon) \sim \mathbf{D}} \frac{1}{2} \left[ 1 - y \mathop{\mathbb{E}}_{h \sim \mathcal{Q}} h(x+\epsilon) \right] \qquad \text{(Markov's ineq. on } y \mathbb{E} h(x+\epsilon)\text{)}.$$

Similarly we have

$$\frac{1}{2}A_{\mathbf{D}^n}(H_{\mathcal{Q}}) = \frac{1}{2} \Pr_{((x,y),\boldsymbol{\mathcal{E}}) \sim \mathbf{D}^n} \left( \exists \epsilon \in \boldsymbol{\mathcal{E}}, y \mathop{\mathbb{E}}_{h \sim \mathcal{Q}} h(x+\epsilon) \leq 0 \right)$$

$$= \frac{1}{2} \Pr_{((x,y),\boldsymbol{\mathcal{E}}) \sim \mathbf{D}^n} \left( \min_{\epsilon \in \boldsymbol{\mathcal{E}}} \left( y \mathop{\mathbb{E}}_{h \sim \mathcal{Q}} h(x+\epsilon) \right) \leq 0 \right)$$

$$= \frac{1}{2} \Pr_{((x,y),\epsilon) \sim \mathbf{D}} \left( 1 - \min_{\epsilon \in \boldsymbol{\mathcal{E}}} \left( y \mathop{\mathbb{E}}_{h \sim \mathcal{Q}} h(x+\epsilon) \right) \geq 1 \right)$$

$$\leq \mathop{\mathbb{E}}_{((x,y),\epsilon) \sim \mathbf{D}} \frac{1}{2} \left[ 1 - \min_{\epsilon \in \boldsymbol{\mathcal{E}}} \left( y \mathop{\mathbb{E}}_{h \sim \mathcal{Q}} h(x+\epsilon) \right) \right] \qquad \text{(Markov's ineq. on } \min y \mathbb{E} h(x+\epsilon)\text{)}.$$

$\square$

# D Proof of Theorem 6

**Theorem 6.** *For any distribution* $\mathbf{D}$ *on* $(X \times Y) \times B$, *for any set of voters* $\mathcal{H}$, *for any prior* $\mathcal{P}$ *on* $\mathcal{H}$, *for any* $n$, *with probability at least* $1-\delta$ *over* $\mathbf{S}$, *for all posteriors* $\mathcal{Q}$ *on* $\mathcal{H}$, *we have*

$$\text{kl}(\overline{R_{\mathbf{S}}}(H_{\mathcal{Q}}) \| \overline{R_{\mathbf{D}}}(H_{\mathcal{Q}})) \leq \frac{1}{m} \left[ \text{KL}(\mathcal{Q} \| \mathcal{P}) + \ln \frac{m+1}{\delta} \right], \tag{14}$$

$$\text{and} \quad \overline{R_{\mathbf{D}}}(H_{\mathcal{Q}}) \leq \overline{R_{\mathbf{S}}}(H_{\mathcal{Q}}) + \sqrt{\frac{1}{2m} \left[ \text{KL}(\mathcal{Q} \| \mathcal{P}) + \ln \frac{m+1}{\delta} \right]}, \tag{15}$$

$$\text{where} \quad \overline{R_{\mathbf{S}}}(H_{\mathcal{Q}}) = \frac{1}{mn} \sum_{i=1}^{m} \sum_{j=1}^{n} \frac{1}{2} \left[ 1 - y_i \mathop{\mathbb{E}}_{h \sim \mathcal{Q}} h(x_i + \epsilon_j^i) \right],$$

$\text{kl}(a \| b) = a \ln \frac{a}{b} + (1-a) \ln \frac{1-a}{1-b}$, *and* $\text{KL}(\mathcal{Q} \| \mathcal{P}) = \mathop{\mathbb{E}}_{h \sim \mathcal{P}} \ln \frac{\mathcal{P}(h)}{\mathcal{Q}(h)}$ *the KL-divergence between* $\mathcal{P}$ *and* $\mathcal{Q}$.

*Proof.* Let $\Gamma = (V, E)$ be the graph representing the dependencies between the random variables where *(i)* the set of vertices is $V = \mathbf{S}$, *(ii)* the set of edges $E$ is defined such that $(((x,y),\epsilon),((x',y'),\epsilon')) \notin E \Leftrightarrow x \neq x'$. Then, applying Th. 8 of Ralaivola et al. [2010] with our notations gives

$$\mathrm{kl}(\overline{R_\mathbf{S}}(H_\mathcal{Q}) \| \overline{R_\mathbf{D}}(H_\mathcal{Q})) \leq \frac{\chi(\Gamma)}{mn} \left[ \mathrm{KL}(\mathcal{Q}\|\mathcal{P}) + \ln \frac{mn + \chi(\Gamma)}{\delta \chi(\Gamma)} \right],$$

where $\chi(\Gamma)$ is the fractional chromatic number of $\Gamma$. From a property of Scheinerman and Ullman [2011], we have

$$c(\Gamma) \leq \chi(\Gamma) \leq \Delta(\Gamma) + 1,$$

where $c(\Gamma)$ is the order of the largest clique in $\Gamma$ and $\Delta(\Gamma)$ is the maximum degree of a vertex in $\Gamma$. By construction of $\Gamma$, $c(\Gamma) = n$ and $\Delta(\Gamma) = n-1$. Thus, $\chi(\Gamma) = n$ and rearranging the terms proves Equation (9). Finally, by applying Pinsker's inequality (*i.e.*, $|a-b| \leq \sqrt{\frac{1}{2} \mathrm{kl}(a\|b)}$), we obtain Equation (10). □

## E  Proof of Theorem 7

**Theorem 7.** *For any distribution* $\mathbf{D}$ *on* $(X \times Y) \times B$, *for any set of voters* $\mathcal{H}$, *for any prior* $\mathcal{P}$ *on* $\mathcal{H}$, *for any* $n$, *with probability at least* $1-\delta$ *over* $\mathbf{S}$, *for all posteriors* $\mathcal{Q}$ *on* $\mathcal{H}$, *for all* $i \in \{1, \dots, m\}$, *for all distributions* $\pi_i$ *on* $\mathcal{E}_i$ *independent from a voter* $h \in \mathcal{H}$, *we have*

$$\overline{A_{\mathbf{D}^n}}(H_\mathcal{Q}) \leq \frac{1}{m} \mathop{\mathbb{E}}_{h \sim \mathcal{Q}} \sum_{i=1}^{m} \max_{\epsilon \in \mathcal{E}_i} \frac{1}{2} (1 - y_i h(x_i + \epsilon)) + \sqrt{\frac{1}{2m} \left[ \mathrm{KL}(\mathcal{Q}\|\mathcal{P}) + \ln \frac{2\sqrt{m}}{\delta} \right]} \quad (16)$$

$$\leq \overline{A_\mathbf{S}}(H_\mathcal{Q}) + \frac{1}{m} \sum_{i=1}^{m} \mathop{\mathbb{E}}_{h \sim \mathcal{Q}} \mathrm{TV}(\rho_i^h \| \pi_i) + \sqrt{\frac{1}{2m} \left[ \mathrm{KL}(\mathcal{Q}\|\mathcal{P}) + \ln \frac{2\sqrt{m}}{\delta} \right]}, \quad (17)$$

*where* $\overline{A_\mathbf{S}}(H_\mathcal{Q}) = \frac{1}{m} \sum_{i=1}^{m} \frac{1}{2} \left[ 1 - \min_{\epsilon \in \mathcal{E}_i} \left( y_i \mathop{\mathbb{E}}_{h \sim \mathcal{Q}} h(x_i + \epsilon) \right) \right]$, *and* $\mathrm{TV}(\rho \| \pi) = \mathop{\mathbb{E}}_{\epsilon \sim \pi} \frac{1}{2} \left| \left[ \frac{\rho(\epsilon)}{\pi(\epsilon)} \right] - 1 \right|$.

*Proof.* Let $L_{h,(x,y),\epsilon} = \frac{1}{2} \left[ 1 - y h(x + \epsilon) \right]$ for the sake of readability. The losses $\max_{\epsilon \in \mathcal{E}_1} L_{h,(x_1,y_1),\epsilon}$, $\dots \max_{\epsilon \in \mathcal{E}_1} L_{h,(x_m,y_m),\epsilon}$ are *i.i.d.* for any $h \in \mathcal{H}$. Hence, we can apply Theorem 20 of Germain et al. [2015] and Pinsker's inequality (*i.e.*, $|q-p| \leq \sqrt{\frac{1}{2} \mathrm{kl}(q\|p)}$) to obtain

$$\mathop{\mathbb{E}}_{h \sim \mathcal{Q}} \mathop{\mathbb{E}}_{(x,y),\mathcal{E}) \sim \mathbf{D}^n} \max_{\epsilon \in \mathcal{E}} L_{h,(x,y),\epsilon} \leq \mathop{\mathbb{E}}_{h \sim \mathcal{Q}} \frac{1}{m} \sum_{i=1}^{m} \max_{\epsilon \in \mathcal{E}_i} L_{h,(x_i,y_i),\epsilon} + \sqrt{\frac{\mathrm{KL}(\mathcal{Q}\|\mathcal{P}) + \ln \frac{2\sqrt{m}}{\delta}}{2m}}.$$

Then, we lower-bound the left-hand side of the inequality with $\overline{A_{\mathbf{D}^n}}(H_\mathcal{Q})$, we have

$$\overline{A_{\mathbf{D}^n}}(H_\mathcal{Q}) \leq \mathop{\mathbb{E}}_{h \sim \mathcal{Q}} \mathop{\mathbb{E}}_{((x,y),\mathcal{E}) \sim \mathbf{D}^n} \max_{\epsilon \in \mathcal{E}} L_{h,(x,y),\epsilon}.$$

Finally, from the definition of $\rho_i^h$, and from Lemma 4 of Ohnishi and Honorio [2021], we have

$$\mathop{\mathbb{E}}_{h \sim \mathcal{Q}} \frac{1}{m} \sum_{i=1}^{m} \max_{\epsilon \in \mathcal{E}_i} L_{h,(x_i,y_i),\epsilon} = \mathop{\mathbb{E}}_{h \sim \mathcal{Q}} \frac{1}{m} \sum_{i=1}^{m} \mathop{\mathbb{E}}_{\epsilon \sim \rho_i^h} L_{h,(x_i,y_i),\epsilon}$$

$$\leq \mathop{\mathbb{E}}_{h \sim \mathcal{Q}} \frac{1}{m} \sum_{i=1}^{m} \mathrm{TV}(\rho_i^h \| \pi_i) + \mathop{\mathbb{E}}_{h \sim \mathcal{Q}} \frac{1}{m} \sum_{i=1}^{m} \mathop{\mathbb{E}}_{\epsilon \sim \pi_i} L_{h,(x_i,y_i),\epsilon}$$

$$= \mathop{\mathbb{E}}_{h \sim \mathcal{Q}} \frac{1}{m} \sum_{i=1}^{m} \mathrm{TV}(\rho_i^h \| \pi_i) + \frac{1}{m} \sum_{i=1}^{m} \mathop{\mathbb{E}}_{\epsilon \sim \pi_i} \mathop{\mathbb{E}}_{h \sim \mathcal{Q}} L_{h,(x_i,y_i),\epsilon}$$

$$\leq \mathop{\mathbb{E}}_{h \sim \mathcal{Q}} \frac{1}{m} \sum_{i=1}^{m} \mathrm{TV}(\rho_i^h \| \pi_i) + \overline{A_\mathbf{S}}(H_\mathcal{Q}).$$

□

## F  Details on the Validity of the Bounds

In this section, we discuss about the validity of the bound when *(i)* generating perturbed sets such as **S** from a distribution **D** dependent on the prior $\mathcal{P}$ *(ii)* selecting the prior $\mathcal{P}$ with $\mathcal{S}_t$.

Actually, computing the bounds implies perturbing examples, *i.e.*, generating examples from **D** that is defined as $\mathbf{D}((x,y),\epsilon) = D(x,y) \cdot \omega_{(x,y)}(\epsilon)$. However, in order to obtain valid bounds, $\omega_{(x,y)}$ must be defined *a priori*. Since the prior $\mathcal{P}$ is defined *a priori* as well, $\omega_{(x,y)}$ can be dependent on $\mathcal{P}$. Hence, $\omega_{(x,y)}$ boils down to generating perturbed example $(x+\epsilon, y)$ by attacking the prior majority vote $H_{\mathcal{P}}$ with PGD$_U$ or IFGSM$_U$. Nevertheless, our selection of the prior $\mathcal{P}$ with $\mathcal{S}$ may seem like "cheating", but this remains a valid strategy when we perform a union bound.

We explain the union bound for Theorem 6, and the same technique can be applied for Theorem 7. Let $\mathbf{D}_1, \dots, \mathbf{D}_T$ be $T$ distributions defined as $\mathbf{D}_1 = D(x,y) \cdot \omega^1_{(x,y)}(\epsilon), \dots, \mathbf{D}_T = D(x,y) \cdot \omega^T_{(x,y)}(\epsilon)$ on $(X \times Y) \times B$ where each distribution $\omega^t_{(x,y)}$ depends on the example $(x,y)$ and possibly on the fixed prior $\mathcal{P}_t$. Furthermore, we denote as $(\mathbf{D}^n_t)^m$ the distribution on the perturbed learning sample consisted of $m$ examples and $n$ perturbations for each example. Then, for all distributions $\mathbf{D}_t$, we can derive a bound on the risk $\overline{R_{\mathbf{D}_t}}(H_{\mathcal{Q}})$ which holds with probability at least $1 - \frac{\delta}{T}$, we have

$$\Pr_{\mathbf{S}_t \sim (\mathbf{D}^n_t)^m}\left[\forall \mathcal{Q},\ \mathrm{kl}(\overline{R_{\mathbf{S}_t}}(H_{\mathcal{Q}})\|\overline{R_{\mathbf{D}_t}}(H_{\mathcal{Q}})) \leq \frac{1}{m}\left[\mathrm{KL}(\mathcal{Q}\|\mathcal{P}_t) + \ln \frac{T(m+1)}{\delta}\right]\right]$$

$$= \Pr_{\mathbf{S}_1 \sim (\mathbf{D}^n_1)^m, \dots, \mathbf{S}_T \sim (\mathbf{D}^n_T)^m}\left[\forall \mathcal{Q},\ \mathrm{kl}(\overline{R_{\mathbf{S}_t}}(H_{\mathcal{Q}})\|\overline{R_{\mathbf{D}_t}}(H_{\mathcal{Q}})) \leq \frac{1}{m}\left[\mathrm{KL}(\mathcal{Q}\|\mathcal{P}_t) + \ln \frac{T(m+1)}{\delta}\right]\right] \geq 1 - \frac{\delta}{T}.$$

Then, from a union bound argument, we have

$$\Pr_{\mathbf{S}_1 \sim (\mathbf{D}^n_1)^m, \dots, \mathbf{S}_T \sim (\mathbf{D}^n_T)^m}\left[\forall \mathcal{Q},\ \mathrm{kl}(\overline{R_{\mathbf{S}_1}}(H_{\mathcal{Q}})\|\overline{R_{\mathbf{D}_1}}(H_{\mathcal{Q}})) \leq \frac{1}{m}\left[\mathrm{KL}(\mathcal{Q}\|\mathcal{P}_t) + \ln \frac{T(m+1)}{\delta}\right],\right.$$

$$\text{and } \dots,$$

$$\left.\text{and } \mathrm{kl}(\overline{R_{\mathbf{S}_T}}(H_{\mathcal{Q}})\|\overline{R_{\mathbf{D}_T}}(H_{\mathcal{Q}})) \leq \frac{1}{m}\left[\mathrm{KL}(\mathcal{Q}\|\mathcal{P}_T) + \ln \frac{T(m+1)}{\delta}\right]\right] \geq 1 - \delta.$$

Hence, we can select $\mathcal{P} \in \{\mathcal{P}_1, \dots, \mathcal{P}_T\}$ with $\mathcal{S}$, and let $\mathbf{D}((x,y),\epsilon) = D(x,y) \cdot \omega_{(x,y)}(\epsilon)$ be the distributions on $(X \times Y) \times B$ where $\omega_{(x,y)}(\epsilon)$ is dependent on $\mathcal{P}$ and on the example $(x,y)$, we can say that

$$\Pr_{\mathbf{S} \sim (\mathbf{D}^n)^m}\left[\forall \mathcal{Q},\ \mathrm{kl}(\overline{R_{\mathbf{S}}}(H_{\mathcal{Q}})\|\overline{R_{\mathbf{D}}}(H_{\mathcal{Q}})) \leq \frac{1}{m}\left[\mathrm{KL}(\mathcal{Q}\|\mathcal{P}) + \ln \frac{T(m+1)}{\delta}\right]\right] \geq 1 - \delta. \tag{18}$$

Additionally, when applying the same process for Equations (11) and (12) in Theorem 7, we have

$$\Pr_{\mathbf{S} \sim (\mathbf{D}^n)^m}\left[\forall \mathcal{Q},\ \overline{A_{\mathbf{D}^n}}(H_{\mathcal{Q}}) \leq \frac{1}{m} \mathbb{E}_{h \sim \mathcal{Q}} \sum_{i=1}^{m} \max_{\epsilon \in \mathcal{E}_i} \frac{1}{2}(1 - y_i h(x_i + \epsilon))\right.$$

$$\left. + \sqrt{\frac{1}{2m}\left[\mathrm{KL}(\mathcal{Q}\|\mathcal{P}) + \ln \frac{2T\sqrt{m}}{\delta}\right]}\right] \geq 1 - \delta, \tag{19}$$

and

$$\Pr_{\mathbf{S} \sim (\mathbf{D}^n)^m}\left[\forall \mathcal{Q},\ \overline{A_{\mathbf{D}^n}}(H_{\mathcal{Q}}) \leq \overline{A_{\mathbf{S}}}(H_{\mathcal{Q}}) + \frac{1}{m}\sum_{i=1}^{m} \mathbb{E}_{h \sim \mathcal{Q}} \mathrm{TV}(\rho^h_i \| \pi_i)\right.$$

$$\left. + \sqrt{\frac{1}{2m}\left[\mathrm{KL}(\mathcal{Q}\|\mathcal{P}) + \ln \frac{2T\sqrt{m}}{\delta}\right]}\right] \geq 1 - \delta. \tag{20}$$

## G  Details on Algorithm 1 and on the computation of the bounds

In this section, we explain how we attack the examples and optimize the bounds in Algorithm 1. Moreover, we present the computation of the bounds after the optimization. Furthermore, remark that the bounds involve the number of epochs $T$ (see Section F for more details).

**Computing the bounds.** Unlike Equation (19) or Equation (20), Equation (18) is not directly optimizable since we upper-bound a deviation (the $\mathrm{kl}$) between the empirical and true risk. Hopefully, we can compute the bound when it is expressed with the inverse binary kl divergence $\mathrm{kl}^{-1}$ defined as $\mathrm{kl}^{-1}(a|\epsilon) = \max_{b \in [0,1]} \{\mathrm{kl}(a\|b) \le \epsilon\}$. Equation (18) can be rewritten as

$$\overline{R_{\mathbf{D}}}(H_{\mathcal{Q}}) \le \mathrm{kl}^{-1}\!\left(\overline{R_{\mathbf{S}}}(H_{\mathcal{Q}})\,\middle|\,\frac{1}{m}\left[\mathrm{KL}(\mathcal{Q}\|\mathcal{P}) + \ln\frac{T(m+1)}{\delta}\right]\right).$$

**Optimizing the bounds.** During each epoch $t$ of Step 2 in Algorithm 1, the posterior distribution $\mathcal{Q}_t$ is updated with $\nabla_{\mathcal{Q}_t} \mathrm{B}_{\mathbb{S}}(H_{\mathcal{Q}_t})$. The objective function associated to Equation (18) of Theorem 6 is

$$\mathrm{B}_{\mathbb{S}}(H_{\mathcal{Q}_t}) = \mathrm{kl}^{-1}\!\left(\overline{R_{\mathbb{S}}}(H_{\mathcal{Q}_t})\,\middle|\,\frac{1}{m}\left[\mathrm{KL}(\mathcal{Q}_t\|\mathcal{P}) + \ln\frac{T(m+1)}{\delta}\right]\right).$$

Note that the derivative of $\mathrm{kl}^{-1}$ and its computation can be found in [Reeb et al., 2018, Appendix A]. On the other hand, the objective function to optimize Equation (20) of Theorem 7 is defined as

$$\mathrm{B}_{\mathbb{S}}(H_{\mathcal{Q}_t}) = \overline{A_{\mathbb{S}}}(H_{\mathcal{Q}_t}) + \sqrt{\frac{1}{2m}\left[\mathrm{KL}(\mathcal{Q}_t\|\mathcal{P}) + \ln\frac{2T\sqrt{m}}{\delta}\right]}.$$

Note that the TV distance is 0 when we sample one noise for each example, *i.e.*, when $n=1$. In consequence, the distance is not optimized in Algorithm 1. However, if we had $n>1$, we would have to minimize it.

**Attacking the examples.** The `attack` function used in Algorithm 1 differs from the attack that generates the perturbed set $\mathbf{S}$ (for the bound). Indeed, at each iteration (in both steps), the function attacks an example with the current model while $\mathbf{S}$ is generated with the prior majority vote $H_{\mathcal{P}}$ (the output of Step 1). Note that for all attacks, in order to be differentiable with respect to the input, we remove the $\mathrm{sign}$ function on the voters' outputs.

## H  About the (differentiable) decision trees

In this section, we introduce the differentiable decision trees, *i.e.*, the voters of our majority vote. Note that we adapt the model of Kontschieder et al. [2016] in order to fit with our framework: a voter must output a real between $-1$ and $+1$. An example of such a tree is represented in Figure 3.

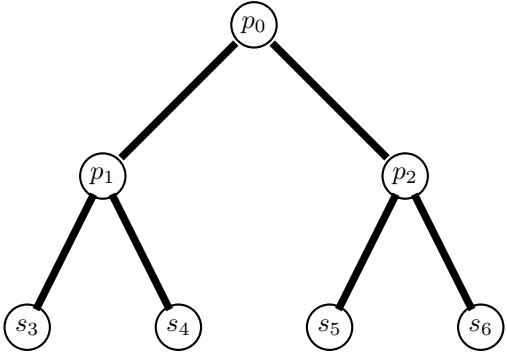

Figure 3: Representation of a (differentiable) decision tree of depth $l = 2$; The root is the node 0 and the leafs are 4; 5; 6 and 7. The probability $p_i(x)$ (respectively $1-p_i(x)$) to go left (respectively right) at the node $i$ is represented by $p_i$ (we omitted the dependence on $x$ for simplicity). Similarly, the predicted label (a "score" between $-1$ and $+1$) at the leaf $i$ is represented by $s_i$.

This differentiable decision tree is stochastic by nature: at each node $i$ of the tree, we continue recursively to the left sub-tree with a probability of $p_i(x)$ and to the right sub-tree with a probability of $1-p_i(x)$; When we attain a leaf $j$, the tree predicts the label $s_j$. Precisely, the probability $p_i(x)$ is constructed by *(i)* selecting randomly 50% of the input features $x$ and applying a random mask $M_i \in \mathbb{R}^d$ on $x$ (where the $k$-th entry of the mask is 1 if the $k$-th feature is selected and 0 otherwise),

by *(ii)* multiplying this quantity by a learned weight vector $v_i \in \mathbb{R}^d$, and by *(iii)* applying a sigmoid function to output a probability. Indeed, we have

$$p_i(x) = \sigma\Big(\langle v_i, M_i \odot x \rangle\Big),$$

where $\sigma(a) = [1 + e^{-a}]^{-1}$ is the sigmoid function; $\langle a, b \rangle$ is the dot product between the vector $a$ and $b$ and $a \odot b$ is the elementwise product between the vector $a$ and $b$. Moreover, $s_i$ is obtained by learning a parameter $u_i \in \mathbb{R}$ and applying a $\tanh$ function, *i.e.*, we have

$$s_i = \tanh\Big(u_i\Big).$$

Finally, instead of having a stochastic voter, $h$ will output the expected label predicted by the tree (see Kontschieder et al. [2016] for more details). It can be computed by $h(x) = f(x, 0, 0)$ with

$$f(x, i, l') = \left\{ \begin{array}{ll} s_i & \text{if } l' = l \\ p_i(x) f(x, 2i{+}1, l'{+}1) + (1 - p_i(x)) f(x, 2i{+}2, l'{+}1) & \text{otherwise} \end{array} \right. .$$

## I  Additional experimental results

In this section, we present the detailed results for the 6 tasks (3 on MNIST and 3 on Fashion MNIST) on which we perform experiments that show the test risks and the bounds for the different scenarios of (Defense, Attack). We train all the models using the same parameters as described in Section 4.2. Table 2 and Table 3 complement Table 1 to present the results for all the tasks when using the $\ell_2$-norm with $b = 1$ (the maximum noise allowed by the norm). Then, we run again the same experiment but we use the $\ell_\infty$-norm with $b = 0.1$ and exhibit the results in Table 4 and Table 5. For the experiments on the 5 other tasks using the $\ell_2$-norm, we have a similar behavior than **MNIST 1vs7** (presented in the paper). Indeed, using the attacks $\text{PGD}_\text{U}$ and $\text{IFGSM}_\text{U}$ as defense mechanism allows to obtain better risks and also tighter bounds compared to the bounds obtained with a defense based on UNIF (which is a naive defense). For the experiments on the 6 tasks using the $\ell_\infty$-norm, the trend is the same as with the $\ell_2$-norm, *i.e.*, the appropriate defense leads to better risks and bounds.

We also run experiments that do not rely on the PAC-Bayesian framework. In other words, we train the models following only Step 1 of our adversarial training procedure (*i.e.*, Algorithm 1) using classical attacks (PGD or IFGSM): we refer to this experiment as a baseline. In our cases, it means learning a majority vote $H_{\mathcal{P}'}$ that follows a distribution $\mathcal{P}'$. As a reminder, the studied scenarios for the baseline are all the pairs (Defense, Attack) belonging to the set $\{\text{—}, \text{UNIF}, \text{PGD}, \text{IFGSM}\} \times \{\text{—}, \text{PGD}, \text{IFGSM}\}$. We report the results in Table 6 and Table 7. With this experiment, we are now able to compare our defense based on $\text{PGD}_\text{U}$ or $\text{IFGSM}_\text{U}$ and a classical defense based on PGD and IFGSM. Hence, considering the test risks $R_{\mathcal{T}}^{\text{ROB}}(H_{\mathcal{Q}})$ (columns "Attack without U" of Tables 1 to 5) and $R_{\mathcal{T}}^{\text{ROB}}(H_{\mathcal{P}'})$ (in Tables 6 and 7) , we observe similar results between the baseline and our framework.

Table 2: Test risks and bounds for 2 tasks of **MNIST** with $n{=}100$ perturbations for all pairs (Defense,Attack) with the two voters' set $\mathcal{H}$ and $\mathcal{H}^{\text{SIGN}}$. The results in **bold** correspond to the best values between results for $\mathcal{H}$ and $\mathcal{H}^{\text{SIGN}}$. To quantify the gap between our risks and the classical definition we put in *italic* the risk of our models against the classical attacks: we replace $\text{PGD}_{\text{U}}$ and $\text{IFGSM}_{\text{U}}$ by $\text{PGD}$ or $\text{IFGSM}$ (*i.e.*, we did *not* sample from the uniform distribution). Since Eq. (12) upperbounds Eq. (11) thanks to the TV term, we compute the two bound values of Theorem 7.

**$\ell_2$-norm, $b=1$ — Algo.1 with Eq. (9) and Algo.1 with Eq. (12)**

| | | $R_{\mathcal{T}}^{\text{ROB}}(H_{\mathcal{Q}})$ (Attack without U) | | $R_{\mathbf{T}}(H_{\mathcal{Q}})$ | | Th. 6 | | $R_{\mathcal{T}}^{\text{ROB}}(H_{\mathcal{Q}})$ (Attack without U) | | $A_{\mathbf{T}}(H_{\mathcal{Q}})$ | | Th. 7 - Eq. (12) | | Th. 7 - Eq. (11) | |
|---|---|---|---|---|---|---|---|---|---|---|---|---|---|---|---|
| Defense | Attack | $\mathcal{H}^{\text{SIGN}}$ | $\mathcal{H}$ | $\mathcal{H}^{\text{SIGN}}$ | $\mathcal{H}$ | $\mathcal{H}^{\text{SIGN}}$ | $\mathcal{H}$ | $\mathcal{H}^{\text{SIGN}}$ | $\mathcal{H}$ | $\mathcal{H}^{\text{SIGN}}$ | $\mathcal{H}$ | $\mathcal{H}^{\text{SIGN}}$ | $\mathcal{H}$ | $\mathcal{H}^{\text{SIGN}}$ | $\mathcal{H}$ |
| — | — | *.015* | *.015* | .015 | .015 | **.060** | .067 | *.015* | *.015* | .015 | .015 | **.129** | 0.135 | **.129** | .135 |
| — | PGD$_\text{U}$ | *.632* | ***.628*** | **.520** | .526 | 1.059 | **.847** | *.672* | ***.641*** | **.683** | .684 | **1.718** | 2.405 | 1.392 | **.962** |
| — | IFGSM$_\text{U}$ | *.447* | ***.443*** | **.157** | .166 | **.387** | .572 | *.461* | ***.451*** | **.337** | .345 | **1.137** | 2.090 | 0.776 | **.669** |
| UNIF | — | *.024* | *.024* | .024 | .024 | **.073** | .083 | *.024* | *.024* | .024 | .024 | **.140** | 0.148 | **.140** | .148 |
| UNIF | PGD$_\text{U}$ | *.646* | ***.619*** | **.486** | .500 | 1.016 | **.809** | *.649* | ***.626*** | **.648** | .650 | **1.646** | 2.417 | 1.338 | **.915** |
| UNIF | IFGSM$_\text{U}$ | *.442* | *.442* | **.128** | .139 | **.316** | .528 | *.442* | *.442* | **.281** | .293 | **.907** | 2.118 | 0.633 | **.617** |
| PGD$_\text{U}$ | — | ***.024*** | *.025* | **.024** | .025 | **.094** | .101 | ***.024*** | *.025* | .024 | .025 | **.158** | 0.163 | **.158** | .163 |
| PGD$_\text{U}$ | PGD$_\text{U}$ | *.148* | ***.135*** | .111 | **.103** | 0.360 | .355 | *.146* | ***.136*** | .129 | **.120** | **.442** | 2.062 | .414 | **.403** |
| PGD$_\text{U}$ | IFGSM$_\text{U}$ | *.104* | ***.103*** | .072 | .072 | 0.277 | .277 | *.102* | *.102* | .090 | **.084** | **.358** | 1.954 | .335 | **.328** |
| IFGSM$_\text{U}$ | — | *.027* | ***.025*** | .027 | **.025** | **.080** | .091 | *.027* | ***.025*** | .027 | .025 | **.146** | 0.154 | **.146** | .154 |
| IFGSM$_\text{U}$ | PGD$_\text{U}$ | *.188* | ***.178*** | .111 | .119 | **.383** | .405 | *.190* | ***.178*** | .126 | .134 | **.501** | 2.063 | .454 | .454 |
| IFGSM$_\text{U}$ | IFGSM$_\text{U}$ | *.126* | ***.115*** | .076 | **.070** | **.248** | .290 | *.127* | ***.115*** | .091 | **.085** | **.371** | 1.918 | **.329** | .342 |

(a) MNIST 4vs9

| | | $R_{\mathcal{T}}^{\text{ROB}}(H_{\mathcal{Q}})$ (Attack without U) | | $R_{\mathbf{T}}(H_{\mathcal{Q}})$ | | Th. 6 | | $R_{\mathcal{T}}^{\text{ROB}}(H_{\mathcal{Q}})$ (Attack without U) | | $A_{\mathbf{T}}(H_{\mathcal{Q}})$ | | Th. 7 - Eq. (12) | | Th. 7 - Eq. (11) | |
|---|---|---|---|---|---|---|---|---|---|---|---|---|---|---|---|
| Defense | Attack | $\mathcal{H}^{\text{SIGN}}$ | $\mathcal{H}$ | $\mathcal{H}^{\text{SIGN}}$ | $\mathcal{H}$ | $\mathcal{H}^{\text{SIGN}}$ | $\mathcal{H}$ | $\mathcal{H}^{\text{SIGN}}$ | $\mathcal{H}$ | $\mathcal{H}^{\text{SIGN}}$ | $\mathcal{H}$ | $\mathcal{H}^{\text{SIGN}}$ | $\mathcal{H}$ | $\mathcal{H}^{\text{SIGN}}$ | $\mathcal{H}$ |
| — | — | *.015* | *.015* | .015 | .015 | **.043** | .045 | *.015* | *.015* | .015 | .015 | **.117** | 0.118 | **.117** | .118 |
| — | PGD$_\text{U}$ | *.279* | ***.271*** | **.232** | .234 | .600 | **.453** | *.284* | ***.274*** | .284 | .284 | **.829** | 1.929 | .724 | **.524** |
| — | IFGSM$_\text{U}$ | *.143* | ***.137*** | **.089** | .090 | **.204** | .227 | *.144* | ***.139*** | **.125** | .127 | **.422** | 1.662 | .337 | **.293** |
| UNIF | — | *.017* | *.017* | .017 | .017 | **.054** | .055 | *.017* | *.017* | .017 | .017 | **.124** | 0.125 | **.124** | .125 |
| UNIF | PGD$_\text{U}$ | *.219* | ***.201*** | **.172** | .177 | .433 | **.350** | *.219* | ***.209*** | **.217** | .218 | **.671** | 1.810 | .565 | **.419** |
| UNIF | IFGSM$_\text{U}$ | *.122* | *.122* | **.052** | .055 | **.119** | .181 | *.122* | *.123* | **.077** | .082 | **.307** | 1.554 | **.242** | .248 |
| PGD$_\text{U}$ | — | ***.013*** | *.015* | **.013** | .015 | .061 | .061 | ***.013*** | *.015* | **.013** | .015 | .131 | 0.130 | .131 | **.130** |
| PGD$_\text{U}$ | PGD$_\text{U}$ | *.057* | *.057* | .045 | **.041** | **.157** | .160 | *.057* | *.057* | .055 | **.045** | **.227** | 1.536 | .218 | .218 |
| PGD$_\text{U}$ | IFGSM$_\text{U}$ | *.043* | *.043* | **.027** | .031 | **.114** | .119 | *.042* | *.043* | .037 | **.035** | **.187** | 1.433 | **.179** | .181 |
| IFGSM$_\text{U}$ | — | *.014* | ***.012*** | .014 | **.012** | .057 | .057 | *.014* | ***.013*** | .014 | **.013** | .128 | **0.127** | .128 | **.127** |
| IFGSM$_\text{U}$ | PGD$_\text{U}$ | *.077* | ***.072*** | .054 | **.043** | **.170** | .174 | *.076* | ***.075*** | .055 | **.052** | **.252** | 1.510 | **.233** | .236 |
| IFGSM$_\text{U}$ | IFGSM$_\text{U}$ | *.055* | ***.048*** | .034 | **.030** | **.105** | .121 | *.052* | ***.051*** | .039 | **.032** | **.191** | 1.379 | **.177** | .185 |

(b) MNIST 5vs6

Table 3: Test risks and bounds for 3 tasks **Fashion MNIST** with $n{=}100$ perturbations for all pairs (Defense,Attack) with the two voters' set $\mathcal{H}$ and $\mathcal{H}^{\text{SIGN}}$. The results in **bold** correspond to the best values between results for $\mathcal{H}$ and $\mathcal{H}^{\text{SIGN}}$. To quantify the gap between our risks and the classical definition we put in *italic* the risk of our models against the classical attacks: we replace $\text{PGD}_U$ and $\text{IFGSM}_U$ by PGD or IFGSM (*i.e.*, we did *not* sample from the uniform distribution). Since Eq. (12) upperbounds Eq. (11) thanks to the TV term, we compute the two bound values of Theorem 7.

**(a) Fashion MNIST Sandall vs Ankle Boot**

| $\ell_2$-norm $b=1$ | | Algo.1 with Eq. (9) — Attack without U $R_\mathcal{T}^{ROB}(H_Q)$ | | $R_\mathbf{T}(H_Q)$ | | Th. 6 | | Algo.1 with Eq. (12) — Attack without U $R_\mathcal{T}^{ROB}(H_Q)$ | | $A_\mathbf{T}(H_Q)$ | | Th. 7 - Eq. (12) | | Th. 7 - Eq. (11) | |
| Defense | Attack | $\mathcal{H}^{\text{SIGN}}$ | $\mathcal{H}$ | $\mathcal{H}^{\text{SIGN}}$ | $\mathcal{H}$ | $\mathcal{H}^{\text{SIGN}}$ | $\mathcal{H}$ | $\mathcal{H}^{\text{SIGN}}$ | $\mathcal{H}$ | $\mathcal{H}^{\text{SIGN}}$ | $\mathcal{H}$ | $\mathcal{H}^{\text{SIGN}}$ | $\mathcal{H}$ | $\mathcal{H}^{\text{SIGN}}$ | $\mathcal{H}$ |
|---|---|---|---|---|---|---|---|---|---|---|---|---|---|---|---|
| — | — | *.021* | ***.020*** | .021 | **.020** | **.060** | 0.070 | *.019* | *.019* | **.019** | **.019** | **.130** | 0.139 | **.130** | 0.139 |
| — | PGD$_U$ | *.695* | ***.650*** | **.494** | .568 | **1.042** | 1.090 | *.677* | *.686* | **.588** | .674 | **1.326** | 2.307 | 1.152 | **1.082** |
| — | IFGSM$_U$ | *.451* | *.451* | **.269** | .328 | **.585** | .731 | *.405* | *.438* | **.295** | .381 | **.878** | 1.971 | **.730** | 0.746 |
| UNIF | — | *.071* | *.071* | .071 | .071 | **.185** | 0.191 | *.071* | *.071* | .071 | .071 | **.236** | 0.241 | **.236** | 0.241 |
| UNIF | PGD$_U$ | ***.423*** | *.477* | **.418** | .425 | 0.957 | **0.755** | *.486* | *.486* | .513 | .513 | **1.372** | 2.173 | 1.151 | **0.869** |
| UNIF | IFGSM$_U$ | ***.326*** | *.331* | .105 | .105 | **.273** | 0.422 | *.333* | *.331* | .144 | **.142** | **.496** | 1.642 | **.397** | 0.504 |
| PGD$_U$ | — | *.034* | ***.032*** | .034 | **.032** | **.094** | 0.114 | *.034* | *.032* | .034 | **.032** | **.158** | 0.174 | **.158** | 0.174 |
| PGD$_U$ | PGD$_U$ | ***.103*** | *.115* | **.086** | .091 | **.227** | 0.289 | *.102* | *.115* | **.096** | .101 | **.299** | 1.985 | **.283** | 0.338 |
| PGD$_U$ | IFGSM$_U$ | ***.092*** | *.099* | **.073** | .076 | **.195** | 0.248 | *.092* | *.099* | .082 | .082 | **.266** | 1.914 | **.253** | 0.299 |
| IFGSM$_U$ | — | ***.028*** | *.030* | .028 | .030 | **.091** | 0.105 | *.027* | *.030* | .027 | .030 | **.155** | 0.166 | **.155** | 0.166 |
| IFGSM$_U$ | PGD$_U$ | *.115* | ***.114*** | .085 | .085 | **.254** | 0.287 | *.112* | *.114* | **.096** | .101 | **.331** | 2.026 | **.313** | 0.337 |
| IFGSM$_U$ | IFGSM$_U$ | *.095* | *.097* | **.067** | .068 | **.206** | 0.232 | *.093* | *.097* | **.080** | .081 | **.282** | 1.927 | **.266** | 0.285 |

**(b) Fashion MNIST Top vs Pullover**

| $\ell_2$-norm $b=1$ | | Algo.1 with Eq. (9) — Attack without U $R_\mathcal{T}^{ROB}(H_Q)$ | | $R_\mathbf{T}(H_Q)$ | | Th. 6 | | Algo.1 with Eq. (12) — Attack without U $R_\mathcal{T}^{ROB}(H_Q)$ | | $A_\mathbf{T}(H_Q)$ | | Th. 7 - Eq. (12) | | Th. 7 - Eq. (11) | |
| Defense | Attack | $\mathcal{H}^{\text{SIGN}}$ | $\mathcal{H}$ | $\mathcal{H}^{\text{SIGN}}$ | $\mathcal{H}$ | $\mathcal{H}^{\text{SIGN}}$ | $\mathcal{H}$ | $\mathcal{H}^{\text{SIGN}}$ | $\mathcal{H}$ | $\mathcal{H}^{\text{SIGN}}$ | $\mathcal{H}$ | $\mathcal{H}^{\text{SIGN}}$ | $\mathcal{H}$ | $\mathcal{H}^{\text{SIGN}}$ | $\mathcal{H}$ |
|---|---|---|---|---|---|---|---|---|---|---|---|---|---|---|---|
| — | — | *.038* | ***.037*** | .038 | **.037** | **.088** | .091 | *.038* | *.037* | .038 | **.037** | **.153** | 0.155 | **.153** | .155 |
| — | PGD$_U$ | *.292* | *.248* | .233 | **.112** | .452 | **.363** | *.289* | *.272* | .287 | **.246** | **.578** | 1.314 | .525 | **.479** |
| — | IFGSM$_U$ | *.194* | *.154* | .132 | **.075** | .300 | **.262** | *.193* | *.181* | .176 | **.148** | **.423** | 1.103 | .376 | **.359** |
| UNIF | — | *.039* | *.039* | **.039** | .039 | **.091** | .093 | *.041* | *.039* | .041 | **.039** | **.155** | 0.157 | **.155** | .157 |
| UNIF | PGD$_U$ | *.240* | *.220* | **.099** | .117 | .346 | **.332** | *.250* | *.231* | .250 | **.245** | **.553** | 1.228 | .490 | **.443** |
| UNIF | IFGSM$_U$ | *.177* | *.171* | **.070** | .078 | **.228** | .247 | *.197* | *.185* | .186 | **.164** | **.445** | 1.046 | .371 | **.346** |
| PGD$_U$ | — | *.045* | *.044* | .045 | **.044** | .108 | **.105** | *.046* | *.045* | .046 | **.045** | .172 | **0.167** | .172 | **.167** |
| PGD$_U$ | PGD$_U$ | *.108* | *.100* | **.077** | .082 | **.203** | .211 | *.104* | *.100* | **.081** | .087 | **.279** | 1.118 | .269 | **.264** |
| PGD$_U$ | IFGSM$_U$ | *.094* | *.086* | .071 | **.069** | **.184** | .186 | *.090* | *.086* | .076 | **.073** | **.257** | 1.015 | .248 | **.241** |
| IFGSM$_U$ | — | ***.041*** | *.043* | **.041** | .043 | **.094** | .101 | *.039* | *.042* | **.039** | .042 | **.158** | 0.163 | **.158** | .163 |
| IFGSM$_U$ | PGD$_U$ | ***.106*** | *.114* | **.078** | .092 | **.220** | .226 | *.109* | *.113* | **.084** | .095 | **.293** | 1.052 | .279 | **.275** |
| IFGSM$_U$ | IFGSM$_U$ | *.082* | *.087* | **.065** | .072 | **.171** | .176 | *.082* | *.089* | **.068** | .078 | **.247** | 0.927 | .234 | **.232** |

**(c) Fashion MNIST Coat vs Shirt**

| $\ell_2$-norm $b=1$ | | Algo.1 with Eq. (9) — Attack without U $R_\mathcal{T}^{ROB}(H_Q)$ | | $R_\mathbf{T}(H_Q)$ | | Th. 6 | | Algo.1 with Eq. (12) — Attack without U $R_\mathcal{T}^{ROB}(H_Q)$ | | $A_\mathbf{T}(H_Q)$ | | Th. 7 - Eq. (12) | | Th. 7 - Eq. (11) | |
| Defense | Attack | $\mathcal{H}^{\text{SIGN}}$ | $\mathcal{H}$ | $\mathcal{H}^{\text{SIGN}}$ | $\mathcal{H}$ | $\mathcal{H}^{\text{SIGN}}$ | $\mathcal{H}$ | $\mathcal{H}^{\text{SIGN}}$ | $\mathcal{H}$ | $\mathcal{H}^{\text{SIGN}}$ | $\mathcal{H}$ | $\mathcal{H}^{\text{SIGN}}$ | $\mathcal{H}$ | $\mathcal{H}^{\text{SIGN}}$ | $\mathcal{H}$ |
|---|---|---|---|---|---|---|---|---|---|---|---|---|---|---|---|
| — | — | *.122* | *.122* | .122 | .122 | **0.276** | 0.286 | *.122* | *.122* | .122 | .122 | **0.318** | 0.328 | **0.318** | 0.328 |
| — | PGD$_U$ | *.744* | *.738* | **.674** | .689 | 1.386 | **1.066** | *.745* | *.740* | **.767** | .768 | **1.773** | 2.386 | 1.576 | **1.180** |
| — | IFGSM$_U$ | *.652* | *.646* | **.454** | .474 | 0.947 | **0.887** | *.659* | *.648* | **.618** | .632 | **1.597** | 2.214 | 1.276 | **0.992** |
| UNIF | — | *.204* | *.204* | .204 | .204 | 0.444 | 0.444 | *.204* | *.204* | .204 | .204 | **0.475** | 0.476 | **0.475** | 0.476 |
| UNIF | PGD$_U$ | *.750* | *.714* | **.682** | .671 | 1.350 | **1.069** | *.750* | *.719* | **.752** | .749 | **1.732** | 2.063 | **1.189** |  |
| UNIF | IFGSM$_U$ | *.605* | *.575* | **.423** | .431 | 0.871 | **0.866** | *.605* | *.578* | **.530** | .526 | **1.304** | 1.860 | 1.091 | **0.956** |
| PGD$_U$ | — | *.168* | *.165* | .168 | **.165** | **0.423** | 0.428 | *.167* | *.165* | .167 | **.165** | 0.463 | **0.461** | 0.463 | **0.460** |
| PGD$_U$ | PGD$_U$ | *.389* | *.402* | .306 | .369 | 0.768 | **0.719** | *.390* | *.402* | **.319** | .403 | **0.847** | 2.354 | 0.810 | **0.755** |
| PGD$_U$ | IFGSM$_U$ | *.361* | *.368* | **.298** | .324 | 0.693 | **0.672** | *.362* | *.368* | **.320** | .361 | **0.799** | 2.258 | 0.754 | **0.707** |
| IFGSM$_U$ | — | *.150* | *.163* | **.150** | .163 | **0.424** | 0.428 | *.149* | *.163* | **.149** | .163 | **0.458** | 0.461 | **0.458** | 0.461 |
| IFGSM$_U$ | PGD$_U$ | ***.391*** | *.428* | .347 | **.292** | 0.778 | **0.757** | *.390* | *.426* | .371 | **.298** | **0.856** | 2.327 | 0.820 | **0.791** |
| IFGSM$_U$ | IFGSM$_U$ | *.356* | *.382* | .291 | **.273** | **0.685** | 0.689 | *.354* | *.382* | .331 | **.278** | **0.772** | 2.218 | 0.734 | **0.723** |

Table 4: Test risks and bounds for 3 tasks of **MNIST** with $n=100$ perturbations for all pairs (Defense,Attack) with the two voters' set $\mathcal{H}$ and $\mathcal{H}^{\text{SIGN}}$. The results in **bold** correspond to the best values between results for $\mathcal{H}$ and $\mathcal{H}^{\text{SIGN}}$. To quantify the gap between our risks and the classical definition we put in *italic* the risk of our models against the classical attacks: we replace PGD$_{\text{U}}$ and IFGSM$_{\text{U}}$ by PGD or IFGSM (*i.e.*, we did *not* sample from the uniform distribution). Since Eq. (12) upperbounds Eq. (11) thanks to the TV term, we compute the two bound values of Theorem 7.

$\ell_\infty$-norm, $b = 0.1$

| | | Algo.1 with Eq. (9) | | | | | | Algo.1 with Eq. (12) | | | | | | | |
| | | Attack without U $R_{\mathcal{T}}^{\text{ROB}}(H_{\mathcal{Q}})$ | | $R_{\mathbf{T}}(H_{\mathcal{Q}})$ | | Th. 6 | | Attack without U $R_{\mathcal{T}}^{\text{ROB}}(H_{\mathcal{Q}})$ | | $A_{\mathbf{T}}(H_{\mathcal{Q}})$ | | Th. 7 - Eq. (12) | | Th. 7 - Eq. (11) | |
| Defense | Attack | $\mathcal{H}^{\text{SIGN}}$ | $\mathcal{H}$ | $\mathcal{H}^{\text{SIGN}}$ | $\mathcal{H}$ | $\mathcal{H}^{\text{SIGN}}$ | $\mathcal{H}$ | $\mathcal{H}^{\text{SIGN}}$ | $\mathcal{H}$ | $\mathcal{H}^{\text{SIGN}}$ | $\mathcal{H}$ | $\mathcal{H}^{\text{SIGN}}$ | $\mathcal{H}$ | $\mathcal{H}^{\text{SIGN}}$ | $\mathcal{H}$ |
|---|---|---|---|---|---|---|---|---|---|---|---|---|---|---|---|
| — | — | *.005* | *.005* | .005 | .005 | **.017** | .019 | *.005* | *.005* | .005 | .005 | **0.099** | 0.100 | **.099** | .100 |
| — | PGD$_{\text{U}}$ | *.454* | *.454* | **.375** | .384 | .770 | **.638** | *.492* | ***.484*** | .480 | **.476** | **1.127** | 2.031 | .946 | **.716** |
| — | IFGSM$_{\text{U}}$ | *.428* | ***.423*** | **.350** | .361 | .727 | **.610** | *.474* | *.465* | .448 | **.443** | **1.061** | 2.008 | .886 | **.686** |
| UNIF | — | *.004* | *.004* | .004 | .004 | **.018** | .019 | *.004* | *.004* | .004 | .004 | **0.099** | 0.100 | **.099** | .100 |
| UNIF | PGD$_{\text{U}}$ | ***.487*** | *.491* | **.369** | .392 | .779 | **.667** | *.512* | ***.507*** | **.484** | .487 | **1.179** | 2.083 | .972 | **.739** |
| UNIF | IFGSM$_{\text{U}}$ | ***.436*** | *.442* | **.325** | .337 | .664 | **.598** | *.466* | *.459* | .417 | .417 | **1.023** | 1.959 | .841 | **.671** |
| PGD$_{\text{U}}$ | — | ***.006*** | *.006* | .006 | .006 | **.024** | .024 | ***.005*** | *.006* | .005 | .006 | .103 | 0.103 | .103 | .103 |
| PGD$_{\text{U}}$ | PGD$_{\text{U}}$ | ***.018*** | *.020* | **.013** | .016 | **.046** | .050 | *.018* | *.020* | .015 | .020 | **0.127** | 1.461 | **.122** | .123 |
| PGD$_{\text{U}}$ | IFGSM$_{\text{U}}$ | ***.020*** | *.021* | **.012** | .016 | **.048** | .054 | *.019* | *.021* | .015 | .020 | **0.130** | 1.455 | **.125** | .127 |
| IFGSM$_{\text{U}}$ | — | ***.006*** | *.007* | **.006** | .007 | **.023** | .024 | *.006* | *.007* | .006 | .007 | **0.102** | 0.103 | **.102** | .103 |
| IFGSM$_{\text{U}}$ | PGD$_{\text{U}}$ | ***.018*** | *.019* | .016 | .016 | **.046** | .051 | *.018* | *.019* | .018 | .019 | **0.126** | 1.489 | **.122** | .124 |
| IFGSM$_{\text{U}}$ | IFGSM$_{\text{U}}$ | *.020* | *.020* | **.015** | .016 | **.050** | .055 | *.020* | *.020* | .020 | **.019** | **0.131** | 1.481 | **.126** | .127 |

(a) MNIST 1 vs 7

| | | Algo.1 with Eq. (9) | | | | | | Algo.1 with Eq. (12) | | | | | | | |
| | | Attack without U $R_{\mathcal{T}}^{\text{ROB}}(H_{\mathcal{Q}})$ | | $R_{\mathbf{T}}(H_{\mathcal{Q}})$ | | Th. 6 | | Attack without U $R_{\mathcal{T}}^{\text{ROB}}(H_{\mathcal{Q}})$ | | $A_{\mathbf{T}}(H_{\mathcal{Q}})$ | | Th. 7 - Eq. (12) | | Th. 7 - Eq. (11) | |
| Defense | Attack | $\mathcal{H}^{\text{SIGN}}$ | $\mathcal{H}$ | $\mathcal{H}^{\text{SIGN}}$ | $\mathcal{H}$ | $\mathcal{H}^{\text{SIGN}}$ | $\mathcal{H}$ | $\mathcal{H}^{\text{SIGN}}$ | $\mathcal{H}$ | $\mathcal{H}^{\text{SIGN}}$ | $\mathcal{H}$ | $\mathcal{H}^{\text{SIGN}}$ | $\mathcal{H}$ | $\mathcal{H}^{\text{SIGN}}$ | $\mathcal{H}$ |
|---|---|---|---|---|---|---|---|---|---|---|---|---|---|---|---|
| — | — | *.015* | *.015* | .015 | .015 | **.060** | .067 | *.015* | *.015* | .015 | .015 | **0.129** | 0.135 | **0.129** | 0.135 |
| — | PGD$_{\text{U}}$ | ***.929*** | *.930* | **.651** | .662 | 1.367 | **1.125** | *.920* | *.925* | **.874** | .880 | **2.213** | 2.661 | 1.792 | **1.266** |
| — | IFGSM$_{\text{U}}$ | *.935* | *.935* | **.601** | .609 | 1.243 | **1.088** | *.926* | *.928* | **.800** | .806 | **2.047** | 2.615 | 1.649 | **1.224** |
| UNIF | — | *.017* | *.017* | .017 | .017 | **.062** | .072 | *.017* | *.017* | .017 | .017 | **0.131** | 0.139 | **0.131** | 0.139 |
| UNIF | PGD$_{\text{U}}$ | *.895* | *.895* | **.615** | .623 | 1.302 | **1.078** | ***.884*** | *.888* | **.815** | .818 | **2.035** | 2.722 | 1.670 | **1.208** |
| UNIF | IFGSM$_{\text{U}}$ | *.898* | *.898* | **.516** | .528 | 1.112 | **1.027** | ***.884*** | *.890* | **.697** | .706 | **1.875** | 2.658 | 1.497 | **1.153** |
| PGD$_{\text{U}}$ | — | *.039* | ***.037*** | .039 | **.037** | **.093** | .094 | *.039* | ***.037*** | .039 | **.037** | **0.156** | 0.157 | **0.156** | 0.157 |
| PGD$_{\text{U}}$ | PGD$_{\text{U}}$ | ***.108*** | *.109* | .090 | .090 | **.200** | .209 | *.108* | *.109* | **.110** | .112 | **0.337** | 1.874 | 0.290 | **0.271** |
| PGD$_{\text{U}}$ | IFGSM$_{\text{U}}$ | ***.121*** | *.124* | **.101** | .103 | **.229** | .235 | *.121* | *.124* | .126 | **.125** | **0.378** | 1.890 | 0.326 | **0.297** |
| IFGSM$_{\text{U}}$ | — | *.046* | ***.044*** | .046 | **.044** | **.102** | .119 | *.046* | ***.044*** | .046 | **.044** | **0.164** | 0.178 | **0.164** | 0.178 |
| IFGSM$_{\text{U}}$ | PGD$_{\text{U}}$ | *.105* | ***.093*** | .091 | **.078** | **.203** | .214 | *.105* | ***.093*** | .108 | **.089** | **0.321** | 1.810 | 0.286 | **0.269** |
| IFGSM$_{\text{U}}$ | IFGSM$_{\text{U}}$ | *.119* | ***.095*** | .102 | **.080** | **.220** | .229 | *.119* | *.095* | .122 | **.090** | **0.357** | 1.821 | 0.309 | **0.283** |

(b) MNIST 4 vs 9

| | | Algo.1 with Eq. (9) | | | | | | Algo.1 with Eq. (12) | | | | | | | |
| | | Attack without U $R_{\mathcal{T}}^{\text{ROB}}(H_{\mathcal{Q}})$ | | $R_{\mathbf{T}}(H_{\mathcal{Q}})$ | | Th. 6 | | Attack without U $R_{\mathcal{T}}^{\text{ROB}}(H_{\mathcal{Q}})$ | | $A_{\mathbf{T}}(H_{\mathcal{Q}})$ | | Th. 7 - Eq. (12) | | Th. 7 - Eq. (11) | |
| Defense | Attack | $\mathcal{H}^{\text{SIGN}}$ | $\mathcal{H}$ | $\mathcal{H}^{\text{SIGN}}$ | $\mathcal{H}$ | $\mathcal{H}^{\text{SIGN}}$ | $\mathcal{H}$ | $\mathcal{H}^{\text{SIGN}}$ | $\mathcal{H}$ | $\mathcal{H}^{\text{SIGN}}$ | $\mathcal{H}$ | $\mathcal{H}^{\text{SIGN}}$ | $\mathcal{H}$ | $\mathcal{H}^{\text{SIGN}}$ | $\mathcal{H}$ |
|---|---|---|---|---|---|---|---|---|---|---|---|---|---|---|---|
| — | — | *.015* | *.015* | .015 | .015 | **.043** | .045 | *.015* | *.015* | .015 | .015 | **0.117** | 0.118 | **0.117** | .118 |
| — | PGD$_{\text{U}}$ | *.500* | ***.499*** | **.387** | .390 | .923 | **.744** | *.502* | ***.500*** | **.474** | .475 | **1.361** | 2.275 | 1.146 | **.830** |
| — | IFGSM$_{\text{U}}$ | *.519* | ***.505*** | **.395** | .398 | .915 | **.762** | ***.514*** | *.516* | .481 | .481 | **1.335** | 2.283 | 1.129 | **.847** |
| UNIF | — | *.015* | *.015* | .015 | .015 | **.052** | .053 | *.015* | *.015* | .015 | .015 | **0.123** | 0.124 | **0.123** | .124 |
| UNIF | PGD$_{\text{U}}$ | ***.529*** | *.544* | **.388** | .393 | .925 | **.761** | ***.517*** | *.532* | .481 | .482 | **1.342** | 2.349 | 1.137 | **.848** |
| UNIF | IFGSM$_{\text{U}}$ | ***.536*** | *.544* | **.372** | .379 | .881 | **.774** | ***.523*** | *.544* | .451 | .456 | **1.268** | 2.348 | 1.077 | **.857** |
| PGD$_{\text{U}}$ | — | *.015* | ***.014*** | .015 | **.014** | **.060** | .064 | *.015* | ***.014*** | .015 | **.014** | **0.130** | 0.133 | **0.130** | .133 |
| PGD$_{\text{U}}$ | PGD$_{\text{U}}$ | *.055* | *.058* | **.037** | .039 | **.131** | .143 | *.056* | *.057* | .046 | .046 | **0.219** | 1.619 | **0.202** | .204 |
| PGD$_{\text{U}}$ | IFGSM$_{\text{U}}$ | ***.061*** | *.065* | **.040** | .043 | **.146** | .154 | ***.059*** | *.062* | .050 | .046 | **0.232** | 1.626 | 0.216 | **.214** |
| IFGSM$_{\text{U}}$ | — | *.019* | ***.014*** | .019 | **.014** | .069 | **.064** | *.018* | ***.014*** | .018 | **.014** | .136 | **0.132** | .136 | **.132** |
| IFGSM$_{\text{U}}$ | PGD$_{\text{U}}$ | *.061* | *.061* | **.040** | .050 | .143 | **.142** | *.061* | *.061* | **.045** | .061 | **0.218** | 1.694 | 0.208 | **.205** |
| IFGSM$_{\text{U}}$ | IFGSM$_{\text{U}}$ | ***.066*** | *.069* | **.044** | .054 | .154 | **.152** | ***.065*** | *.069* | .048 | .068 | **0.228** | 1.708 | 0.216 | **.214** |

(c) MNIST 5 vs 6

Table 5: Test risks and bounds for 3 tasks of **Fashion MNIST** with $n=100$ perturbations for all pairs (Defense,Attack) with the two voters' set $\mathcal{H}$ and $\mathcal{H}^{\text{SIGN}}$. The results in **bold** correspond to the best values between results for $\mathcal{H}$ and $\mathcal{H}^{\text{SIGN}}$. To quantify the gap between our risks and the classical definition we put in *italic* the risk of our models against the classical attacks: we replace $\text{PGD}_{\text{U}}$ and $\text{IFGSM}_{\text{U}}$ by PGD or IFGSM (*i.e.*, we did *not* sample from the uniform distribution). Since Eq. (12) upperbounds Eq. (11) thanks to the TV term, we compute the two bound values of Theorem 7.

**(a) Fashion MNIST Sandall vs Ankle Boot** — $\ell_\infty$-norm, $b=0.1$

| Defense | Attack | $R^{\text{ROB}}_{\mathcal{T}}(H_{\mathcal{Q}})$ $\mathcal{H}^{\text{SIGN}}$ | $\mathcal{H}$ | $R_{\mathbf{T}}(H_{\mathcal{Q}})$ $\mathcal{H}^{\text{SIGN}}$ | $\mathcal{H}$ | Th. 6 $\mathcal{H}^{\text{SIGN}}$ | $\mathcal{H}$ | $R^{\text{ROB}}_{\mathcal{T}}(H_{\mathcal{Q}})$ $\mathcal{H}^{\text{SIGN}}$ | $\mathcal{H}$ | $A_{\mathbf{T}}(H_{\mathcal{Q}})$ $\mathcal{H}^{\text{SIGN}}$ | $\mathcal{H}$ | Th. 7 - Eq. (12) $\mathcal{H}^{\text{SIGN}}$ | $\mathcal{H}$ | Th. 7 - Eq. (11) $\mathcal{H}^{\text{SIGN}}$ | $\mathcal{H}$ |
|---|---|---|---|---|---|---|---|---|---|---|---|---|---|---|---|
| — | — | *.021* | ***.020*** | .021 | **.020** | **0.060** | 0.070 | *.019* | *.019* | .019 | .019 | **0.130** | 0.139 | **0.130** | 0.139 |
| — | PGD$_\text{U}$ | *.951* | ***.944*** | **.606** | .719 | **1.275** | 1.333 | *.935* | *.920* | **.762** | .864 | **1.617** | 2.503 | 1.421 | **1.317** |
| — | IFGSM$_\text{U}$ | *.957* | *.947* | **.588** | .718 | **1.231** | 1.336 | *.950* | *.950* | **.734** | .851 | **1.587** | 2.495 | 1.395 | **1.316** |
| UNIF | — | ***.076*** | *.077* | **.076** | .077 | **0.178** | 0.184 | *.076* | *.077* | **.076** | .077 | **0.230** | 0.235 | **0.230** | 0.235 |
| UNIF | PGD$_\text{U}$ | *.964* | *.961* | **.714** | .719 | 1.496 | **1.265** | *.966* | *.963* | **.853** | .859 | 2.098 | 2.417 | 1.785 | **1.416** |
| UNIF | IFGSM$_\text{U}$ | *.978* | *.976* | **.627** | .632 | 1.306 | **1.259** | *.979* | *.979* | **.758** | .762 | 1.914 | 2.422 | 1.597 | **1.396** |
| PGD$_\text{U}$ | — | *.041* | *.040* | .041 | **.040** | 0.114 | **0.111** | *.041* | *.040* | .041 | **.040** | 0.173 | **0.171** | 0.173 | **0.171** |
| PGD$_\text{U}$ | PGD$_\text{U}$ | *.098* | ***.097*** | .089 | **.086** | **0.207** | 0.210 | *.099* | *.097* | .101 | **.100** | **0.306** | 1.826 | 0.281 | **0.267** |
| PGD$_\text{U}$ | IFGSM$_\text{U}$ | *.113* | ***.112*** | .105 | **.101** | **0.244** | 0.246 | *.115* | *.112* | .120 | **.113** | **0.353** | 1.853 | 0.321 | **0.302** |
| IFGSM$_\text{U}$ | — | ***.045*** | *.047* | .045 | **.047** | **0.131** | 0.137 | *.045* | *.047* | .045 | .047 | **0.188** | 0.194 | **0.188** | 0.194 |
| IFGSM$_\text{U}$ | PGD$_\text{U}$ | ***.100*** | *.102* | .089 | **.085** | **0.203** | 0.232 | *.100* | *.102* | .102 | .102 | **0.298** | 1.645 | 0.274 | **0.287** |
| IFGSM$_\text{U}$ | IFGSM$_\text{U}$ | ***.112*** | *.116* | .099 | **.096** | **0.232** | 0.260 | *.112* | *.116* | .114 | **.112** | **0.328** | 1.687 | 0.301 | **0.313** |

**(b) Fashion MNIST Top vs Pullover** — $\ell_\infty$-norm, $b=0.1$

| Defense | Attack | $R^{\text{ROB}}_{\mathcal{T}}(H_{\mathcal{Q}})$ $\mathcal{H}^{\text{SIGN}}$ | $\mathcal{H}$ | $R_{\mathbf{T}}(H_{\mathcal{Q}})$ $\mathcal{H}^{\text{SIGN}}$ | $\mathcal{H}$ | Th. 6 $\mathcal{H}^{\text{SIGN}}$ | $\mathcal{H}$ | $R^{\text{ROB}}_{\mathcal{T}}(H_{\mathcal{Q}})$ $\mathcal{H}^{\text{SIGN}}$ | $\mathcal{H}$ | $A_{\mathbf{T}}(H_{\mathcal{Q}})$ $\mathcal{H}^{\text{SIGN}}$ | $\mathcal{H}$ | Th. 7 - Eq. (12) $\mathcal{H}^{\text{SIGN}}$ | $\mathcal{H}$ | Th. 7 - Eq. (11) $\mathcal{H}^{\text{SIGN}}$ | $\mathcal{H}$ |
|---|---|---|---|---|---|---|---|---|---|---|---|---|---|---|---|
| — | — | *.038* | ***.037*** | .038 | **.037** | **.088** | .091 | *.038* | *.037* | .038 | **.037** | **0.153** | 0.155 | **0.153** | .155 |
| — | PGD$_\text{U}$ | *.596* | ***.515*** | .477 | **.218** | .844 | **.662** | *.590* | *.576* | .570 | **.502** | **1.049** | 1.924 | 0.948 | **.857** |
| — | IFGSM$_\text{U}$ | *.723* | ***.623*** | .573 | **.257** | .971 | **.751** | *.716* | *.695* | .678 | **.598** | **1.189** | 2.031 | 1.080 | **.980** |
| UNIF | — | *.032* | *.032* | .032 | .032 | **.083** | .085 | *.032* | *.033* | .032 | .033 | **0.149** | 0.151 | **0.149** | .151 |
| UNIF | PGD$_\text{U}$ | ***.438*** | *.439* | .356 | **.245** | .813 | **.563** | *.435* | *.435* | .423 | **.312** | **1.082** | 1.867 | 0.959 | **.688** |
| UNIF | IFGSM$_\text{U}$ | ***.546*** | *.547* | .453 | **.325** | .974 | **.690** | *.544* | *.547* | .530 | **.409** | **1.266** | 2.009 | 1.128 | **.823** |
| PGD$_\text{U}$ | — | ***.048*** | *.053* | **.048** | .053 | **.115** | .130 | *.048* | *.053* | **.048** | .053 | **0.177** | 0.188 | **0.177** | .188 |
| PGD$_\text{U}$ | PGD$_\text{U}$ | ***.102*** | *.116* | .089 | **.099** | **.205** | .223 | *.102* | *.116* | .096 | **.115** | **0.282** | 1.323 | 0.266 | .278 |
| PGD$_\text{U}$ | IFGSM$_\text{U}$ | ***.120*** | *.135* | .102 | **.115** | **.237** | .255 | *.120* | *.135* | .109 | **.133** | **0.318** | 1.380 | 0.299 | .309 |
| IFGSM$_\text{U}$ | — | *.051* | ***.045*** | .051 | **.045** | .120 | **.115** | *.051* | *.045* | .051 | **.045** | 0.179 | **0.175** | 0.179 | **.175** |
| IFGSM$_\text{U}$ | PGD$_\text{U}$ | *.106* | ***.094*** | .091 | **.085** | .211 | **.193** | *.106* | *.094* | .102 | **.097** | **0.292** | 1.488 | 0.273 | **.252** |
| IFGSM$_\text{U}$ | IFGSM$_\text{U}$ | *.120* | ***.111*** | **.101** | .102 | .239 | **.218** | *.119* | *.111* | .113 | .113 | **0.322** | 1.546 | 0.299 | **.277** |

**(c) Fashion MNIST Coat vs Shirt** — $\ell_\infty$-norm, $b=0.1$

| Defense | Attack | $R^{\text{ROB}}_{\mathcal{T}}(H_{\mathcal{Q}})$ $\mathcal{H}^{\text{SIGN}}$ | $\mathcal{H}$ | $R_{\mathbf{T}}(H_{\mathcal{Q}})$ $\mathcal{H}^{\text{SIGN}}$ | $\mathcal{H}$ | Th. 6 $\mathcal{H}^{\text{SIGN}}$ | $\mathcal{H}$ | $R^{\text{ROB}}_{\mathcal{T}}(H_{\mathcal{Q}})$ $\mathcal{H}^{\text{SIGN}}$ | $\mathcal{H}$ | $A_{\mathbf{T}}(H_{\mathcal{Q}})$ $\mathcal{H}^{\text{SIGN}}$ | $\mathcal{H}$ | Th. 7 - Eq. (12) $\mathcal{H}^{\text{SIGN}}$ | $\mathcal{H}$ | Th. 7 - Eq. (11) $\mathcal{H}^{\text{SIGN}}$ | $\mathcal{H}$ |
|---|---|---|---|---|---|---|---|---|---|---|---|---|---|---|---|
| — | — | *.122* | *.122* | .122 | .122 | **0.276** | 0.286 | *.122* | *.122* | .122 | .122 | **0.318** | 0.328 | **0.318** | 0.328 |
| — | PGD$_\text{U}$ | ***.884*** | *.887* | **.781** | .795 | 1.579 | **1.268** | *.882* | *.886* | **.864** | .872 | **2.020** | 2.640 | 1.803 | **1.390** |
| — | IFGSM$_\text{U}$ | ***.901*** | *.902* | **.756** | .774 | 1.558 | **1.272** | *.901* | *.902* | **.865** | .876 | **2.032** | 2.651 | 1.795 | **1.393** |
| UNIF | — | *.166* | *.166* | .166 | .166 | **0.352** | 0.357 | *.166* | *.166* | .166 | .166 | **0.389** | 0.394 | **0.389** | 0.394 |
| UNIF | PGD$_\text{U}$ | *.911* | *.914* | **.796** | .798 | 1.402 | **1.326** | *.913* | *.914* | .896 | **.888** | **1.934** | 2.325 | 1.713 | **1.447** |
| UNIF | IFGSM$_\text{U}$ | *.935* | *.937* | **.787** | .798 | 1.392 | **1.350** | *.934* | *.936* | .887 | **.882** | **1.905** | 2.378 | 1.693 | **1.469** |
| PGD$_\text{U}$ | — | *.163* | ***.162*** | .163 | **.162** | **0.386** | 0.395 | *.163* | *.162* | .163 | **.162** | **0.419** | 0.430 | **0.419** | 0.430 |
| PGD$_\text{U}$ | PGD$_\text{U}$ | *.394* | *.396* | .359 | **.329** | 0.764 | **0.673** | *.394* | *.396* | .403 | **.394** | **0.954** | 2.321 | 0.865 | **0.726** |
| PGD$_\text{U}$ | IFGSM$_\text{U}$ | *.475* | *.480* | .442 | **.410** | 0.910 | **0.769** | *.477* | *.480* | .487 | **.472** | **1.121** | 2.411 | 1.020 | **0.826** |
| IFGSM$_\text{U}$ | — | *.167* | *.168* | **.167** | .168 | 0.411 | **0.395** | *.167* | *.168* | .167 | .168 | 0.445 | **0.429** | 0.445 | **0.429** |
| IFGSM$_\text{U}$ | PGD$_\text{U}$ | *.396* | ***.373*** | .359 | **.293** | 0.772 | **0.641** | *.396* | *.373* | .405 | **.328** | **0.970** | 2.368 | 0.877 | **0.692** |
| IFGSM$_\text{U}$ | IFGSM$_\text{U}$ | *.465* | ***.428*** | .424 | **.334** | 0.891 | **0.705** | *.465* | *.429* | .470 | **.372** | **1.090** | 2.425 | 0.995 | **0.758** |

Table 6: Test risks for 6 tasks of **MNIST** and **Fashion MNIST** datasets for all pairs (Defense,Attack) with the two voters' set $\mathcal{H}$ and $\mathcal{H}^{\text{SIGN}}$ using $\ell_2$-norm. The results of these tables are computed considering defenses of the literature, *i.e.*, adversarial training using PGD or IFGSM. We also add an adversarial training using UNIF for the completeness of comparison between this baseline defense and our algorithm. The results in **bold** correspond to the best values between results for $\mathcal{H}$ and $\mathcal{H}^{\text{SIGN}}$.

| $\ell_2$-norm, $b=1$ | | $R_\mathcal{T}^{\text{ROB}}(H_{\mathcal{P}'})$ | |
|---|---|---|---|
| Defense | Attack | $\mathcal{H}^{\text{SIGN}}$ | $\mathcal{H}$ |
| — | — | .005 | .005 |
| — | PGD | **.326** | .327 |
| — | IFGSM | .122 | **.121** |
| UNIF | — | .005 | .005 |
| UNIF | PGD | .191 | **.190** |
| UNIF | IFGSM | **.071** | .072 |
| PGD | — | .007 | .007 |
| PGD | PGD | .027 | **.026** |
| PGD | IFGSM | .022 | **.021** |
| IFGSM | — | **.005** | .006 |
| IFGSM | PGD | .041 | **.035** |
| IFGSM | IFGSM | .021 | .021 |

(a) MNIST 1 vs 7

| $\ell_2$-norm, $b=1$ | | $R_\mathcal{T}^{\text{ROB}}(H_{\mathcal{P}'})$ | |
|---|---|---|---|
| Defense | Attack | $\mathcal{H}^{\text{SIGN}}$ | $\mathcal{H}$ |
| — | — | .015 | .015 |
| — | PGD | .692 | .692 |
| — | IFGSM | .464 | **.462** |
| UNIF | — | .024 | .024 |
| UNIF | PGD | .653 | .653 |
| UNIF | IFGSM | .441 | **.438** |
| PGD | — | **.024** | .027 |
| PGD | PGD | **.136** | .138 |
| PGD | IFGSM | **.097** | .102 |
| IFGSM | — | **.022** | .027 |
| IFGSM | PGD | **.166** | .186 |
| IFGSM | IFGSM | **.113** | .124 |

(b) MNIST 4 vs 9

| $\ell_2$-norm, $b=1$ | | $R_\mathcal{T}^{\text{ROB}}(H_{\mathcal{P}'})$ | |
|---|---|---|---|
| Defense | Attack | $\mathcal{H}^{\text{SIGN}}$ | $\mathcal{H}$ |
| — | — | .015 | .015 |
| — | PGD | .283 | .283 |
| — | IFGSM | .144 | .144 |
| UNIF | — | .017 | .017 |
| UNIF | PGD | .220 | **.219** |
| UNIF | IFGSM | .122 | .122 |
| PGD | — | .014 | **.013** |
| PGD | PGD | .056 | **.055** |
| PGD | IFGSM | .045 | **.041** |
| IFGSM | — | **.013** | .014 |
| IFGSM | PGD | .077 | **.070** |
| IFGSM | IFGSM | .053 | **.047** |

(c) MNIST 5 vs 6

| $\ell_2$-norm, $b=1$ | | $R_\mathcal{T}^{\text{ROB}}(H_{\mathcal{P}'})$ | |
|---|---|---|---|
| Defense | Attack | $\mathcal{H}^{\text{SIGN}}$ | $\mathcal{H}$ |
| — | — | .019 | .019 |
| — | PGD | .709 | **.708** |
| — | IFGSM | .426 | **.414** |
| UNIF | — | **.071** | .072 |
| UNIF | PGD | .531 | .531 |
| UNIF | IFGSM | .331 | **.329** |
| PGD | — | **.034** | .036 |
| PGD | PGD | .107 | **.103** |
| PGD | IFGSM | .091 | **.087** |
| IFGSM | — | .031 | **.029** |
| IFGSM | PGD | .125 | **.108** |
| IFGSM | IFGSM | .104 | **.090** |

(d) Fashion MNIST
Sandall vs Ankle Boot

| $\ell_2$-norm, $b=1$ | | $R_\mathcal{T}^{\text{ROB}}(H_{\mathcal{P}'})$ | |
|---|---|---|---|
| Defense | Attack | $\mathcal{H}^{\text{SIGN}}$ | $\mathcal{H}$ |
| — | — | .038 | .038 |
| — | PGD | .286 | **.285** |
| — | IFGSM | .188 | **.186** |
| UNIF | — | .041 | **.039** |
| UNIF | PGD | .249 | **.248** |
| UNIF | IFGSM | .197 | **.192** |
| PGD | — | **.043** | .045 |
| PGD | PGD | **.102** | .117 |
| PGD | IFGSM | **.090** | .094 |
| IFGSM | — | **.038** | .040 |
| IFGSM | PGD | .120 | **.106** |
| IFGSM | IFGSM | .092 | **.080** |

(e) Fashion MNIST
Top vs Pullover

| $\ell_2$-norm, $b=1$ | | $R_\mathcal{T}^{\text{ROB}}(H_{\mathcal{P}'})$ | |
|---|---|---|---|
| Defense | Attack | $\mathcal{H}^{\text{SIGN}}$ | $\mathcal{H}$ |
| — | — | .122 | .122 |
| — | PGD | .768 | **.767** |
| — | IFGSM | .683 | **.680** |
| UNIF | — | .204 | .204 |
| UNIF | PGD | **.753** | .754 |
| UNIF | IFGSM | .607 | **.606** |
| PGD | — | .182 | **.178** |
| PGD | PGD | .453 | **.412** |
| PGD | IFGSM | .408 | **.379** |
| IFGSM | — | .148 | **.146** |
| IFGSM | PGD | **.405** | .411 |
| IFGSM | IFGSM | .369 | **.364** |

(f) Fashion MNIST
Coat vs Shirt

Table 7: Test risks for 6 tasks of **MNIST** and **Fashion MNIST** datasets for all pairs (Defense,Attack) with the two voters' set $\mathcal{H}$ and $\mathcal{H}^{\text{SIGN}}$ using $\ell_\infty$-norm. The results of these tables are computed considering defenses of the literature, *i.e.*, adversarial training using PGD or IFGSM. We also add an adversarial training using UNIF for the completeness of comparison between this baseline defense and our algorithm. The results in **bold** correspond to the best values between results for $\mathcal{H}$ and $\mathcal{H}^{\text{SIGN}}$.

| $\ell_\infty$-norm, $b=0.1$ | | $R^{\text{ROB}}_{\mathcal{T}}(H_{\mathcal{P}'})$ | |
|---|---|---|---|
| Defense | Attack | $\mathcal{H}^{\text{SIGN}}$ | $\mathcal{H}$ |
| — | — | .005 | .005 |
| — | PGD | .499 | **.498** |
| — | IFGSM | **.479** | .480 |
| UNIF | — | .004 | .004 |
| UNIF | PGD | .516 | **.515** |
| UNIF | IFGSM | .467 | .467 |
| PGD | — | **.006** | .007 |
| PGD | PGD | .019 | .019 |
| PGD | IFGSM | .021 | .021 |
| IFGSM | — | .007 | .007 |
| IFGSM | PGD | **.017** | .018 |
| IFGSM | IFGSM | **.019** | .020 |

(a) MNIST 1 vs 7

| $\ell_\infty$-norm, $b=0.1$ | | $R^{\text{ROB}}_{\mathcal{T}}(H_{\mathcal{P}'})$ | |
|---|---|---|---|
| Defense | Attack | $\mathcal{H}^{\text{SIGN}}$ | $\mathcal{H}$ |
| — | — | .015 | .015 |
| — | PGD | .921 | .921 |
| — | IFGSM | .923 | .923 |
| UNIF | — | .017 | .017 |
| UNIF | PGD | .877 | **.876** |
| UNIF | IFGSM | .877 | .877 |
| PGD | — | .041 | **.040** |
| PGD | PGD | **.108** | .109 |
| PGD | IFGSM | **.122** | .123 |
| IFGSM | — | .057 | **.044** |
| IFGSM | PGD | .109 | **.101** |
| IFGSM | IFGSM | .119 | **.108** |

(b) MNIST 4 vs 9

| $\ell_\infty$-norm, $b=0.1$ | | $R^{\text{ROB}}_{\mathcal{T}}(H_{\mathcal{P}'})$ | |
|---|---|---|---|
| Defense | Attack | $\mathcal{H}^{\text{SIGN}}$ | $\mathcal{H}$ |
| — | — | .015 | .015 |
| — | PGD | .498 | .498 |
| — | IFGSM | .511 | **.510** |
| UNIF | — | .015 | .015 |
| UNIF | PGD | .512 | **.511** |
| UNIF | IFGSM | .511 | .511 |
| PGD | — | .014 | .014 |
| PGD | PGD | .065 | **.058** |
| PGD | IFGSM | .068 | **.065** |
| IFGSM | — | .018 | **.017** |
| IFGSM | PGD | **.061** | .063 |
| IFGSM | IFGSM | **.069** | .071 |

(c) MNIST 5 vs 6

| $\ell_\infty$-norm, $b=0.1$ | | $R^{\text{ROB}}_{\mathcal{T}}(H_{\mathcal{P}'})$ | |
|---|---|---|---|
| Defense | Attack | $\mathcal{H}^{\text{SIGN}}$ | $\mathcal{H}$ |
| — | — | .019 | .019 |
| — | PGD | .938 | .938 |
| — | IFGSM | **.948** | .949 |
| UNIF | — | **.076** | .077 |
| UNIF | PGD | .970 | **.969** |
| UNIF | IFGSM | .981 | .981 |
| PGD | — | .041 | **.040** |
| PGD | PGD | .098 | **.097** |
| PGD | IFGSM | .115 | **.111** |
| IFGSM | — | .112 | **.047** |
| IFGSM | PGD | **.045** | .100 |
| IFGSM | IFGSM | **.101** | .114 |

(d) Fashion MNIST
Sandall vs Ankell Boot

| $\ell_\infty$-norm, $b=0.1$ | | $R^{\text{ROB}}_{\mathcal{T}}(H_{\mathcal{P}'})$ | |
|---|---|---|---|
| Defense | Attack | $\mathcal{H}^{\text{SIGN}}$ | $\mathcal{H}$ |
| — | — | .038 | .038 |
| — | PGD | **.574** | .577 |
| — | IFGSM | .700 | **.696** |
| UNIF | — | **.032** | .033 |
| UNIF | PGD | **.428** | .435 |
| UNIF | IFGSM | **.540** | .550 |
| PGD | — | **.047** | .049 |
| PGD | PGD | .101 | **.097** |
| PGD | IFGSM | .118 | **.112** |
| IFGSM | — | .049 | **.048** |
| IFGSM | PGD | .100 | **.090** |
| IFGSM | IFGSM | .112 | **.108** |

(e) Fashion MNIST
Top vs Pullover

| $\ell_\infty$-norm, $b=0.1$ | | $R^{\text{ROB}}_{\mathcal{T}}(H_{\mathcal{P}'})$ | |
|---|---|---|---|
| Defense | Attack | $\mathcal{H}^{\text{SIGN}}$ | $\mathcal{H}$ |
| — | — | .122 | .122 |
| — | PGD | .879 | .879 |
| — | IFGSM | .898 | .898 |
| UNIF | — | .166 | .166 |
| UNIF | PGD | .913 | **.911** |
| UNIF | IFGSM | .934 | **.933** |
| PGD | — | **.164** | .167 |
| PGD | PGD | .398 | **.395** |
| PGD | IFGSM | **.479** | .481 |
| IFGSM | — | **.163** | .169 |
| IFGSM | PGD | **.356** | .391 |
| IFGSM | IFGSM | **.422** | .461 |

(f) Fashion MNIST
Coat vs Shirt