# OpenReview forum: "A PAC-Bayes Analysis of Adversarial Robustness"
_NeurIPS.cc/2021/Conference — NeurIPS 2021 Poster_

### Official Review · Reviewer_Jhjk · 2021-07-15

**Rating:** 4
**Confidence:** 4

**Summary:**

This paper studies the problem of robustness to evasion attacks from a learning theory point of view. Unlike most existing works from the literature, the authors do not study the uniform bounds on the worst-case risk under attack (a.k.a. adversarial risk). Instead they introduce a new analytical framework that considers less powerful randomized adversaries and majority vote classifiers.  This new setting allows them to leverage existing results from the PAC-Bayes literature; hence providing generalization bounds on the averaged risk under attacks of a majority vote classifier (where the average is taken over the randomness of the adversary).

**Limitations And Societal Impact:**

I don’t see any possibly negative social impact of this work.

**Main Review:**

Relevance: The topic of evasion attacks is still lacking some theoretical ground. Accordingly, presenting new theoretical results to this community is an interesting and relevant line of research.

Clarity: The paper is well written and understandable. Nevertheless I feel like some statements (listed below) lack justification or at least were not perfectly clear to me.
- In line 44. The authors say that their definition has the advantage of being suited to majority voting. While I agree with this statement, it is unclear to me why previous definitions were not. Indeed, I can always define my hypothesis space as being a set of classifiers that have been constructed by majority voting, which allows me to apply every classical definition from adversarial learning theory (including adversarial risk). Did I misunderstood something here?
- In line 46. The authors claim that their bounds are independent from the kinds of attacks they consider. However, it seems to me that the attack they consider does have an impact in the bound presented in Theorem 7. Indeed the TV term depends on $\mathcal{E}_i$ which is directly sampled from $D$ (i.e., the distribution characterizing the set of randomized attacks considered).
- In lines 208-209. It was a bit unclear to me what the TV term in Theorem 7 stands for. The authors state that this term goes to $0$ when the voters are fooled by the same attack $\epsilon \in \mathcal{E}_i$, but such a setting seems counterproductive to me. If all the classifiers are fooled by the same attack, then I don’t see how building majority vote classifiers with them will bring robustness to the system.
- In line 249, the authors present their randomized adaptation of $PGD$ and $IFGSM$ attacks, namely $PGD_U$ and $IFGSM_U$. I felt like the definition of these new attacks was a bit confusing. More precisely, it was a bit unclear to me whether the uniform noise was injected before performing the attack or after (i.e., directly adversarial example computed with $PGD$ or $IFGSM$). I think it is the latter, but to make this clearer, the author could present these attacks a bit more formally (only one line explaining where the uniform noise is injected).

Quality: The technical part of the paper looks sound to me. The main issue I have with this part is that it does not clearly relate to the problem of adversarial examples usually considered by the ML community. In general, in the problem we are interested in (even for a fixed model and not a class of models), we study a worst-case scenario, i.e., if the attacker can find the solution to (5). Here the authors consider a much weaker model where the attacker can only select the best attack from a fixed-size set of random points. While the authors claim that their definition can be connected to the classical problem (Propositions 3 and 4), this connection seems quite loose to me and not very informative. Indeed, as there is no concrete values for $TV(\Pi,\Delta)$, the left-hand term in (8) could be arbitrarily small. I think it would have been much more convincing to present a connection that depends on the number of samples one considers, not only an inequality like in Proposition 3, but a true dynamic depending on $n$ and $D$.

Similarly, I think the experimental part of the paper was not convincingly illustrating the connection between the adversarial risk and the averaged risks the paper introduces. The authors are only studying an empirical version of the adversarial risk, that has been calculated using a specific attack (that might as well be suboptimal). Accordingly, I don’t think that the author should claim to provide non-vacuous bound on the adversarial risk. On the randomized attack part, if my understanding of the adversarial attack $PGD_U$ and $IFGSM_U$ is right, I think these attacks are suboptimal. In fact, there is no clear reason to me why injecting uniform noise on an existing attack could provide stronger examples. To consider randomized attacks in a stronger scenario, I suggest that the author uses random restart instead (i.e., inject the uniform noise before computing the attack) which is a well-known technique to improve the efficiency of PGD and IFGSM. More generally, the authors could strengthen the experimental protocol by following some good practices to guarantee that the attacks have been correctly computed (see e.g. [1]).

Originality: To the best of my knowledge, this work is the first one to consider adversarial robustness through the lens of the PAC-Bayes Learning. Nevertheless, I feel like the authors failed to compare with (or at least to discuss) some important lines of research that are connected to the present work.
- The classifiers considered in the paper are built by a majority voting rule as $\text{sign}(E_{h\sim\mathcal{Q} }[h(x)])$. This formulation is very similar to certifiable defenses based on randomized smoothing (see e.g. [3,4]). Let us for example consider a real-valued hypothesis $h$, and $h_\delta$ the function defined as $h_\delta(x) := h(x+\delta)$. Let us also set $\mathcal{Q}$ to be the distribution that samples $\delta$ from a Gaussian distribution $\mathcal{N}(0,\sigma^2)$ and outputs $h_\delta$. Then $\text{sign}(E_{h\sim\mathcal{Q}}[h(x)])$ reduces to applying randomized smoothing with Gaussian noise $\mathcal{N}(0,\sigma^2)$. Accordingly, it would be interesting if the author could explain how these techniques differ from theirs, if their bounds are applicable to randomly smoothed classifiers and how their bounds can be compared to the certificates presented in previous works.
- One of the originality of this paper is to consider randomized attackers. In this context, it could be interesting to compare to some recent theoretical works (see e.g. [4,5] ) investigating adversarial robustness from a distributional point of view (i.e., where the attacker can be random).

[1] Florian Tramèr, Nicholas Carlini, Wieland Brendel, Aleksander Madry. On Adaptive Attacks to Adversarial Example Defenses. NeurIPS 2020.

[2] Jeremy Cohen, Elan Rosenfeld, and Zico Kolter. Certified adversarial robustness via randomized smoothing. ICML 2019.

[3] Hadi Salman, Greg Yang, Jerry Li, Pengchuan Zhang, Huan Zhang, Ilya Razenshteyn, Sebastien Bubeck. Provably Robust Deep Learning via Adversarially Trained Smoothed Classifiers. NeurIPS 2020.

[4] Muni Sreenivas Pydi, Varun Jog. Adversarial Risk via Optimal Transport and Optimal Couplings. ICML 2020.

[5] Laurent Meunier, Meyer Scetbon, Rafael Pinot, Jamal Atif, Yann Chevaleyre. Mixed Nash Equilibria in the Adversarial Examples Game. ICML 2021.


**Time Spent Reviewing:**

6 to 8 hours

---

> ### Author Response · Authors · 2021-08-09
> **Answer to reviewer Jhjk**
>
> Thank you for your review and useful comments. Thank you also for pointing out the typo and the ambiguity we might have introduced, we will correct our paper.
>
> About line 44 (... their definition has the advantage of being suited to majority voting ...), you are right, the “averaged(-max)” and “worst-case” definitions  are applicable to majority votes. However, contrary to the worst case definition, we are averaging perturbed examples instead of considering only one worst-case perturbed example.
> We wanted to highlight that despite this averaging we are still suited to majority vote.
> We will clarify it in our paper, we propose : “These definitions, based on averaged quantities on averaged quantities  , have the advantages (i) of still being suitable for the PAC-Bayesian framework and majority vote classifiers, …”
>
> About Line 46 (… their bounds are independent of the kinds of attacks they consider. …), the term “independent” is indeed not the right term to use here. We should rather say “generalize”. Indeed, our formulation allows generating bound values for any kind of attack. We will modify the sentence as follows “Then, we derive a PAC-Bayesian generalization bound for each of our adversarial risks that have the advantage to generalize to any kind of attacks”.
>
> About lines 208 and 209, it is better to think about Theorem 7 in terms of a trade-off between the empirical risk, which reflects the robustness of the majority vote, and two “complexity” terms: the KL term and the TV term. The KL term will control how much the posterior weights of the majority vote can differ from the prior ones. The TV term controls, in some sense, the diversity of the voters, i.e., the ability of the voters to be fooled on the same adversarial example. We will precise it in the text to avoid confusion.
>
> About line 249, the attack PGDU and IFGSMU are variants of the attack PGD and IFGSM. Indeed, let $(x, y)$ be a non-adversarial example, the idea to obtain an attacked example $(x+\epsilon, y)$ consists in
> (i) attacking the prior majority vote $H_{\cal P}$ with the original attack PGD or IFGSM: we will obtain a first perturbation $\epsilon_{1}$
> (ii) sampling a uniform noise which is our perturbation $\epsilon_{2}$
> (iii) then, we set $\epsilon=\epsilon_{1}+\epsilon_{2}$
> We will make it more explicit in the paper.
>
> Concerning using random restart instead of injecting uniform noises, is a good suggestion (thank you for that). Indeed, it can bring new insights on the behavior of the approach and also to consider other realistic scenarios. However, our primary goal was to show that our framework can be applied to provide a generalization bound on sampled noises. Thus, your suggestion can be a natural extension to enhance our first work. We will then mention this in our perspectives.
>
> In [2], they get a certification of robustness. However, their analysis holds only for a fixed model which is in contrast with our analysis that holds for all models. Also, they used a gaussian data augmentation to train their model in order to still have a good performance when predicting on perturbed examples. First of all, we have a similar setting: we specialized the model to weighted majority votes while their approach works for an arbitrary model and they specialized their perturbations to Gaussian noises while our definition of noises is more general. Note that we can also use Gaussian noises. Secondly,  the certificate in [2] consists in saying what is the minimum noise to get an adversarial example while our bounds give us a probability to be fooled by an adversarial example. Furthermore, our learning is made to optimize our bound while in [2] the learning is not related to the certificate.
> Similar remarks apply for [3].
> Thank you for pointing out the references [4,5]. These works have connections to our contribution by the notion of stochastic models as you mentioned and the guarantees provided correspond to bounds on adversarial risks. As far as we can see, the theoretical settings seem a bit different: by the way the attacks are defined and by the use of specific frameworks such as optimal transport or game theory. It is not obvious that the different results are directly comparable, this requires some specific additional work. We can mention anyway that in both papers, the bounds are not necessarily tight: this is discussed in the conclusion of [4], and the use of Rademacher complexity in addition to some potential high constants in [5] will make the bound potentially loose, note that none of these papers show experimental evaluation of the proposed upper bounds. The considered settings do not allow to optimize directly the proposed bounds.
> We will cite the references [1-5] in the related work accordingly and mention the associated perspectives.

---

### Official Review · Reviewer_N9WH · 2021-07-15

**Rating:** 6
**Confidence:** 3

**Summary:**

This paper provides a PAC-Bayes analysis of adversarial robustness. In opposite to standard techniques that derive a worst-case analysis of the risk over all the possible perturbations, the authors use PAC-Bayes bounds to design a learning algorithm that minimizes an adversarial risk. They also provide numerical evidence to support their theoretical results.


**Limitations And Societal Impact:**

No specific societal impact.

**Main Review:**

This paper is clear and well written, and the theoretical analysis is sound, and technically quite solid. These are nice results. I enjoyed the idea of using PAC-Bayes to analyze adversarial robustness and obtain theoretical guarantees on the adversarial risk. However, I'm not convinced that in general, considering the averaged-risk can really help much for adversarial attack as it is generally a classifier- and sample-dependent risk.

I enjoyed the experimental section as well. However, a point that I do not understand (and that I think quite important as the aim of PAC-Bayes is to provide a learning algorithm through the bound) is why the authors do not optimize the TV distance in the bound. They say in the supplementary material that "this term is 0 when we sample one noise for each example", but this is not a requirement, and the TV distance is a key quantity in the bounds of the authors, specific to their analysis. Also, I would have liked seeing experiments on more complicated datasets like CIFAR-10.

**Time Spent Reviewing:**

10

---

> ### Author Response · Authors · 2021-08-09
> **Answer to reviewer N9WH**
>
> First of all, we would like to thank you for your review and all the useful comments.
>
> About your concern: “However, I'm not convinced that in general, considering the averaged-risk can really help much for adversarial attack as it is generally a classifier- and sample-dependent risk.”, we recall that we provide a guarantee on the averaged(-max) adversarial risk which is related to the “worst-case” adversarial risk by Propositions 3 and 4. Our study, which, again, comes with a guarantee, could be seen as complementary to worst-case analyses which can be non-informative like the Rademacher complexity-based one. Indeed, to the best of our knowledge our bounds are the first that are non-vacuous in the context of adversarial training. Hence, considering the averaged risk can help to understand “worst-case” adversarial robustness.
>
> About the optimization of the TV distance in the bound: you are right, in the general case, the TV distance is not 0 and should be considered in a learning procedure as future work.
> However, we make the algorithmic choice to get rid of the TV through an algorithmic procedure that leads to a TV equals zero: we draw only one perturbed example by iteration.

---

> > ### Comment · Reviewer_N9WH · 2021-08-24
> > **Re**
> >
> > I thank the authors for their detailed reply and for their nice work. After careful consideration, I have decided to leave my score unchanged.

---

### Official Review · Reviewer_y2sg · 2021-07-16

**Rating:** 6
**Confidence:** 3

**Summary:**

The paper studies robust generalization using the PAC-Bayes framework. Specifically, the authors establish bounds on relaxations of the worst-case adversarial risk. In the first relaxation (Theorem 6), they are concerned with the average-risk with respect to a distribution over adversarial distributions (instead of worst-case). The second relaxation (Theorem 7), is concerned with an average-max risk, where instead of drawing a single adversarial perturbation, n adversarial perturbations are drawn and then the worst-case among them is counted. The authors also provide a bound on the gap between their relaxations and the worst-case adversarial risk (Proposition), which depends on the total-variation distance between the fixed distribution and the worst-case.

The authors also ran empirical experiments with an adversarial learning procedure that is inspired by the theoretical bounds.

**Limitations And Societal Impact:**

In my opinion, the authors have adequately addressed the limitations of their work. Also, while it is true that the work is mainly theoretical, the authors should address how their theoretical guarantees can impact society.


**Main Review:**

Applying Bac-Bayes analysis to obtain robust generalization bounds is new and hasn’t been considered in prior work to the best of my knowledge. However, what is concerning is the fact that the bounds established are on relaxations of the worst-case adversarial risk.

The authors do not provide strong justification for why such relaxations are considered, e.g., is it the case that the analysis breaks down when studying the worst-case adversarial risk?

While the relaxations may still be useful, I am not totally convinced of that yet. For example, the total-variation distance in Proposition 4 can be arbitrarily large, and will really depend on how good the choice of distribution D is. Even though the authors empirically show that the gap is small on datasets of MNIST and Fashion-MNIST, I am not sure if these are appropriate datasets to compare the gap between the relaxations and the worst-case since they are in a sense "easy" datasets that we can train robust models against.

Can the results in Theorems 6 and 7 be extended to obtain generalization guarantees against a specific attacking algorithm? For example, what if in distribution D the adversarial perturbations are generated by the PGD attack. The challenge that I see is that in this case the distribution over adversarial perturbations will depend on the specific predictor/classifier that is being attacked. But this is really what is sort of happening in the experiments in Section 4. Where a prior P_0 is chosen, then distribution D over adversarial perturbations is determined as a function of this prior (and PGD attack), then a posterior Q_0 is learned, then this is repeated again with prior P_1=Q_0, and so on. It would be nice if the theoretical guarantees in Theorems 6 and 7 can be extended to capture this dynamic.

In Section 4.2, I believe that including the simple baseline of adversarial training with a single classifier (compared to a majority-vote) would be useful. Also, how is the majority-vote predictor attacked? It would be good to include details of this.

-------------------------------------------------------------------------------------------------------------------------------------------------------------------------------------

After Rebuttal.

I thank the authors for their response. After consideration, I decided to keep my score the same.
I do think that this would be a stronger contribution if perhaps the authors could strengthen their results in Theorems 6 and 7 to obtain generalization guarantees against specific attacking algorithms. This would be one way to show case the usefulness of considering the PAC-Bayes framework, and the relaxations to adversarial risk considered in this work.

While the authors comment in their response that this is possible, I am not fully sure that it follows immediately, and perhaps some care is needed.

**Time Spent Reviewing:**

5

---

> ### Author Response · Authors · 2021-08-09
> **Answer to reviewer y2sg**
>
> We would like to thank you for your review and useful comments.
> We will correct all the unclear points and typos pointed out.
>
>
> First, the use of the relaxation instead of the worst-case adversarial risk helps to fully leverage the PAC-Bayesian framework. Indeed, the PAC-Bayesian framework is better suited for analyzing average cases and not pure worst cases. The idea is to provide a novel analysis with the PAC-Bayes framework: the first results using the relaxation are promising and both definitions (relaxation and worst case) are related through Proposition 3 and 4. Thus it is interesting to analyze the relaxations whose results may be useful to bound the worst-case adversarial risk (Figure 1).
>
>
> You are right about the total variation: the value of the distance may be arbitrarily large.
> As you stated, the TV distance depends on the choice of the perturbed examples’ distribution ${\bf D}$. Hence, the TV distance will be large when the chance to draw an adversarial example from ${\bf D}$ is low. To reduce this effect, we define the distribution with respect to the attack PGDU or IFGSMU in order to have a higher chance to sample an adversarial example from ${\bf D}$. This allows us to be closer to an adversarial example thanks to the “specific attacking algorithm” (worst-case) and thus to reduce the value of the TV distance.
> Furthermore, the principle of PGDU and IFGSMU is to apply a “worst-case” attack (either PGD or IFGSM) on an example to get a first perturbed example and then sample a “noisy” version of this perturbed example by drawing a noise from the uniform distribution.
>
>
> About “extending” Theorems 6 and 7 for “specific attacking algorithm”: it is possible to obtain generalization guarantees against a specific attacking algorithm. For example, we can apply the attack PGD in Algorithm 1. Then, the “perturbations in ${\bf D}$” are obtained by running PGD with multiple random starts (each random start will give another perturbation).
> Finally, the defined setting allows us to learn a majority vote using Algorithm 1; it remains to compute the bounds as stated in our paper. Note we can only attack the prior network with PGD in order to obtain a valid PAC-Bayesian guarantee.
>
>
> Concerning the simple baseline of adversarial training (compared to a majority vote). Tables 6 and 7 in Appendix I show the results on a majority vote classifier with a training procedure that does not rely on the PAC-Bayesian framework. More specifically, we used a “classical” adversarial training (using PGD or IFGSM): this experiment corresponds to an adversarial training baseline for majority votes.  We did not use a single classifier to be fair in the comparison of the result because majority votes would be more expressive than a single classifier. We wanted to highlight the impact of the optimization given by the PAC-Bayesian analysis on the relaxed definitions compared to the robust optimization using adversarial training that relies on either PGD or IFGSM.
> The majority vote predictor is attacked in the following way. We used two attacks, PGDU and IFGSMU (as explained above):
> (i) During the training (for Algorithm 1), we attack the current model by finding a perturbation (using PGDU or IFGSMU) for each example in the mini-batch at each iteration of the learning phase.
> (ii) At test time we attack the prior majority-vote predictor $H_{\cal P}$ with either PGDU or IFGSMU on the test set ${\cal T}$ to obtain the perturbed test set ${\bf T}$.
> (iii) At the computation of the bound, we proceed in the same way but with the training set ${\cal S}$ to obtain ${\bf S}$.

---

### Official Review · Reviewer_KAbu · 2021-07-23

**Rating:** 6
**Confidence:** 4

**Summary:**

The paper studies the adversarial robustness of the majority vote method by deriving PAC-Bayesian generalization bounds in the adversarial settings.
In details,  the adversarial robustness means that an algorithm can still work well when the input covariates "X" is adding some noise but not the labels "Y".


**Limitations And Societal Impact:**

Yes

**Main Review:**

The paper, itself, introduces two novel definitions for (and relates to) the classical adversarial risk, which are "Averaged adversarial risk" and "Averaged-max adversarial risk".
PAC-Bayesian empirical bounds are derived with respect to these two risks for the "surrogates linear loss" of the weighted majority vote.
An algorithm is also given by minimizing these empirical bounds and some applications to real data sets are given to illustrate the soundness.

Although the used Pac-Bayesian bound technique in this paper is known, the settings considered is novel. Thus, I believe these theoretical results are an important contribution.

However, my main concern is that:
As in Theorem 6, the authors noted that the result does not depend on "n".
While Theorem 7 is stated for all 'n', So, could you compare these bounds?

Other comments:
1. (line 15-17) any reference for this claim? "This phenomenon known as adversarial robustness contributes to the impossibility to ensure the safety of machine learning algorithms for safety-critical applications such as aeronautics functions (e.g., vision-based navigation), autonomous driving or medical diagnosis."
2. (line 19) is badly written, I can not understand this "...of its input; We talk about adversarial examples. In other words, ......."
3. (line 74) "The learner objective is then to...." should be "The learner's objective....."

**Time Spent Reviewing:**

5

---

> ### Author Response · Authors · 2021-08-09
> **Answer to reviewer KAbu**
>
> First, we would like to thank you for your review and useful comments.
> We will correct all the unclear points and typos pointed out (points 2 & 3). Indeed, we propose to modify line 19 into “Adversarial robustness is thus a critical issue in machine learning that studies the ability of a model to be robust or invariant to perturbations of its input. A perturbed input is usually called an adversarial example.”
>
>
> About the main concern (i.e., about Theorem 6 that does not depend on $n$ and about the comparison between Theorems 6 and 7), we show that $n$ does not have an influence on the bound of Theorem 6 (ie the right-hand side of Equation (9)). However, $n$ is nevertheless present in the empirical risk $\overline{R_{\bf S}(H_{\cal Q})}$. Actually, the bound of Theorem 7 is a pessimistic version of the bound of Theorem 6. In Theorem 7, we take the worst perturbed example among the $n$ generated examples (i.e. the one that worsens the risk). We will clarify this point in the paper.
>
>
> About the first comment (“any reference for this claim”), we propose to add a reference to a survey (see [1]) about the safety and trustworthiness of machine learning systems.
>
> [1] A Survey of Safety and Trustworthiness of Deep Neural Networks: Verification, Testing, Adversarial Attack and Defence, and Interpretability, https://arxiv.org/abs/1812.08342, 2018.

---

### Author Response · Authors · 2021-08-09
**General answer to reviewers**

We appreciate the positive feedbacks highlighting the novelty of our work in the context of adversarial learning -the first to provide a PAC-Bayesian analysis dedicated to it-, the soundness of the technical results and the quality of the paper. We thank reviewer Jhjk for evaluating our work as an interesting and relevant line of research, reviewer KAbu for finding that our theoretical results represent an important contribution and reviewer N9WH for mentioning that our results are nice. We hope that this work can be helpful to the community by presenting a novel point of view on the problem. We are very grateful for the numerous feedbacks and suggestions that helped to improve our work. We address the few concerns and provide answers to specific questions for each review.

---

### Decision · Program_Chairs · 2021-09-28

**Decision:**

Accept (Poster)

**Comment:**

The author(s) provide the 1st PAC-Bayes bounds proving robust generalization in the context of adversarial robustness. The risk they consider is not the usual "worst-case adversarial risk", but randomized versions (or relaxations), that imply a less powerful adversary. Two versions are considered: Theorem 6 is a bound on an average risk with respect to the distribution on the adversary, while Theorem 7 considers the worst case of n random attacks.

The paper relies on standard PAC-Bayes techniques, but they are applied to a new setting: adversarial robustness. Given the lack of theoretical results in this setting, all Reviewers agree that such results are potentially very interesting.

On the other hand, the relaxations studied in Theorem 6 and Theorem 7 are not standard in the literature on adversarial robustness. Three Reviewers: y2sg, Jhjk and N9WH, questioned these definitions and expressed doubt that they would be of interest to researchers in the field of adversarial robustness. The discussion allowed some clarifications, but overall, it remains clear that, while these results are new, the authors failed to convince that they will be useful for the community of adversarial robustness.

For this reason, I recommend to reject the paper. I encourage the author(s) to work on a new version of the paper, but you should either work on the worst-case risk, or explain to what extend the average risks they introduce is related to the worst-case risk. You should also take into account the typos found by Reviewer KAbu and the many comments of Reviewer Jhjk. I wish you good luck with the revision of the paper.

**Consistency Experiment:**

NeurIPS has a long history of experimentation. In 2014, NeurIPS ran an experiment in which 10% of submissions were reviewed by two independent committees to quantify the randomness in the review process. This year, we repeated a variant of this experiment to see how the quality of the review process has changed over time.  This paper was part of the experiment and was therefore assigned to two committees (consisting of reviewers, an Area Chair, and a Senior Area Chair) that reached independent decisions.  If both committees made the same recommendation, this recommendation was followed. If a single committee recommended acceptance, the paper was accepted (with the exception of a few cases in which the other committee identified what we considered a fatal flaw, e.g., an error in a key result).

This copy’s committee reached the following decision: **Reject**

The other committee assigned to the paper recommended **Accept (Poster)**.  You can find the other set of reviews, along with any follow up discussion with the authors here:
https://openreview.net/forum?id=sUBSPowU3L5